# Gene therapy conversion of striatal astrocytes into GABAergic neurons in mouse models of Huntington's disease

Zheng Wu [1], Matthew Parry [1], Xiao-Yi Hou[1], Min-Hui Liu[2], Hui Wang[1,3], Rachel Cain[1], Zi-Fei Pei[1], Yu-Chen Chen[1], Zi-Yuan Guo[1], Sambangi Abhijeet [1] & Gong Chen[1,2 ✉]

Huntington's disease (HD) is caused by Huntingtin (Htt) gene mutation resulting in the loss of striatal GABAergic neurons and motor functional deficits. We report here an in vivo cell conversion technology to reprogram striatal astrocytes into GABAergic neurons in both R6/2 and YAC128 HD mouse models through AAV-mediated ectopic expression of NeuroD1 and Dlx2 transcription factors. We found that the astrocyte-to-neuron (AtN) conversion rate reached 80% in the striatum and >50% of the converted neurons were DARPP32$^+$ medium spiny neurons. The striatal astrocyte-converted neurons showed action potentials and synaptic events, and projected their axons to the targeted globus pallidus and substantia nigra in a time-dependent manner. Behavioral analyses found that NeuroD1 and Dlx2-treated R6/2 mice showed a significant extension of life span and improvement of motor functions. This study demonstrates that in vivo AtN conversion may be a disease-modifying gene therapy to treat HD and other neurodegenerative disorders.

[1] Department of Biology, Huck Institutes of Life Sciences, Pennsylvania State University, University Park, PA 16802, USA. [2] Guangdong-Hong Kong-Macau Institute of CNS Regeneration, Jinan University, Guangzhou, China. [3] Department of Neurology, Affiliated ZhongDa Hospital, School of Medicine, Southeast University, Nanjing, China. ✉email: gongchenpsu@yahoo.com

Huntington's disease (HD) is an autosomal dominant disease characterized by progressive motor, cognitive, and psychiatric symtpoms associated with neuronal dysfunction and atrophy of the striatum and other brain regions[1]. HD arises due to an expanded chain of polyglutamines in the N terminal region of the huntingtin protein (Htt) leading to intracellular accumulation and aggregation of mutant HTT (mHtt)[2,3]. The GABAergic medium spiny neurons (MSNs), which represent about 95% of the total neurons within the human striatum[4], are susceptible to mHtt aggregates and consequently show early degeneration in the striatum of HD patients[5]. Many of the symptoms observed in HD patients are highly related to the loss of MSNs in the striatum[6] and can be mimicked in rodent[7], pig[8], and non-human primate[9] models. Previous studies have achieved some success in HD animal models through cell transplantation[10–12] or using antisense oligonucleotides[13,14] and CRISPR/Cas9 gene editing technology[15] to reduce the mHtt level. Tetrabenazine has been used to treat chorea in HD and other movement disorders but with significant side effects[16].

Recent in vivo cell conversion technology provides an alternative approach to regenerate functional new neurons in adult mammalian brains by directly reprogramming local glial cells into neurons[17–23]. Using endogenous glial cells for neuroregeneration has emerged as a cell replacement therapy that would avoid transplantation of external cells and therefore minimize potential immunorejection caused by foreign cells[24–26]. Glial cells are the most abundant cell type in adult mammalian brains, constituting over half of the total brain cells. In contrast to non-dividing neurons, glial cells have intrinsic proliferative capability and therefore can serve as source cells for regenerative purpose. In vivo direct reprogramming of adult glial cells into functional neurons has achieved repeated success in both astrocytes and NG2 cells in the mouse brain[17–19,21,22,27,28] and even monkey brains[29]. To date, while MSNs have been reprogrammed from fibroblast cells in vitro[30], direct in vivo conversion of striatal glial cells into MSNs has not been achieved.

In this study, we discovered that striatal astrocytes can be directly converted into MSNs in the R6/2 mouse model by coexpression of two neural transcription factors, NeuroD1 and Dlx2, through AAV-based gene therapy. Brain slice recordings reveal that the astrocyte-converted neurons can fire repetitive action potentials and display spontaneous synaptic events, indicating that they are electrophysiologically functional and forming synaptic circuits with other neurons. Interestingly, the converted neurons also can project their axon nerve terminals to distant targets such as substantia nigra pars reticulata (SNr) and the external globus pallidus (GP). Most importantly, in vivo regeneration of GABAergic neurons in the striatum reduces the striatum atrophy, and alleviates a series of phenotypic abnormalities including the improvement of motor functions in the R6/2 HD mouse model. Therefore, our studies provide the proof-of-principle that it is possible to regenerate MSNs in the striatum using local glial cells for the treatment of HD.

## Results

**Targeting striatal astrocytes for in vivo neuronal conversion.** Astrocytes are abundant cells that make up ~30% of the cells in the mammalian CNS and essentially surround every single neuron in the brain[31], making them an ideal internal source for cell conversion. Our previous study has demonstrated that ectopic expression of a single neural transcription factor, NeuroD1, in cortical astrocytes can convert them into functional neurons[18]. However, the total number of in vivo converted neurons by retroviruses is limited, because retroviruses can only express target genes in dividing cells. To overcome this disadvantage of retroviruses, we have made recombinant adeno-associated virus (serotype 2/5, rAAV2/5) for in vivo reprogramming. Among different serotypes of rAAV, we chose rAAV2/5 because it has been reported to infect astrocytes preferentially in the mouse brain[32].

To track the astrocyte-converted neurons in the mouse brain, we developed a Cre-FLEx (flip-excision) system which includes a vector expressing Cre recombinase under the control of the GFAP promoter (GFAP::Cre) to target astrocytes, and FLEx vectors with an inverted coding sequence of mCherry-P2A-mCherry or NeuroD1-P2A-mCherry or Dlx2-P2A-mCherry (Fig. 1a). The two inserted genes are separated by P2A self-cleavage site and driven by the strong universal synthetic promoter CAG. Because Dlx2 is a transcription factor critical for generating GABAergic neurons[33,34,30,35,36], we tested whether Dlx2 in combination with NeuroD1, which mainly generate glutamatergic neurons[18], can convert striatal astrocytes into GABAergic neurons.

To test whether Cre recombinase is specifically overexpressed in the astrocytes, we first injected AAV2/5 GFAP::Cre into the normal mouse striatum (2–5 months), a brain region enriched with GABAergic neurons, which shows early degeneration in HD brains. As expected, almost all of the Cre-expressing cells were GFAP-positive cells, a typical marker for astrocytes ($99.2 \pm 0.6\%$, $n = 6$ mice, 7–21 days post viral injection; Fig. 1b). In order to further investigate the specificity of our Cre-FLEx system, we injected AAV2/5 GFAP::Cre together with AAV2/5-CAG::FLEx-mCherry-P2A-mCherry into the normal mouse striatum. The mice were sacrificed at 21–30 days post-injection (dpi) for immunohistological studies. Of the mCherry-positive cells, the majority of them expressed astrocyte-specific markers including S100β ($90.0 \pm 0.9\%$), GFAP ($86.6 \pm 1.9\%$) and glutamine synthetase (GS, $92.9 \pm 1.3\%$), with very few expressing other glial markers such as Olig2 ($1.1 \pm 0.3\%$), NG2 ($3.2 \pm 1.5\%$), and Iba1 (not detected, $n \geq 6$ mice for each group; Fig. 1c, d). A few mCherry-expressing cells were NeuN-positive ($10.5 \pm 0.7\%$, $n = 11$ mice; Fig. 1c, d), indicating that a small number of striatal neurons were targeted by our AAV2/5 Cre-FLEx system.

**NeuroD1 and Dlx2 reprogram striatal astrocytes into GABAergic neurons.** We next tested whether our AAV Cre-FLEx system could drive the conversion of astrocytes into neurons in the striatum by injecting AAV2/5 GFAP::Cre together with AAV2/5-CAG::Dlx2-P2A-mCherry and CAG::NeuroD1-P2A-mCherry into adult wild-type (WT) mice (age 2–5 months). At 7 dpi, we found that all the viral infected cells (mCherry[+]) in the striatum were GFAP[+] astrocytes, among which 81.5% of the mCherry[+] cells also coexpressed NeuroD1 (ND1) and Dlx2, while only 12.1% of the mCherry[+] cells showed neither ND1 nor Dlx2 expression (Fig. 2a, quantified in Fig. 2c). A small percentage (<5%) of the mCherry[+] cells expressed only one of the two transcriptional factors (either ND1[+] or Dlx2[+]), but neither of the TFs was detected in NeuN[+] neurons at 7 dpi (Fig. 2a and Supplementary Fig. 1a). In contrast, by 30 dpi, we found that most of the ND1 and Dlx2 signals were coexpressed in NeuN[+] neurons (72.7%; Fig. 2b, quantified in Fig. 2c, black dots; Supplementary Fig. 1b), with only a small percentage in astrocytes (4.1%, Fig. 2c, gray dots). These results suggest that coexpression of NeuroD1 and Dlx2 can convert striatal astrocytes into neurons (Fig. 2d).

To further investigate the time course of the astrocyte-to-neuron conversion process in the striatum, three more time points of 11, 15, and 21 dpi were analyzed in addition to 7 and 30 dpi (Fig. 2e). We found that a small percentage (17.8%) of mCherry[+] cells showed NeuN[+] signal after coexpressing NeuroD1 plus Dlx2 (N + D) at 11 dpi, and such neuronal conversion percentage continuously increased to 33.6% at 15 dpi

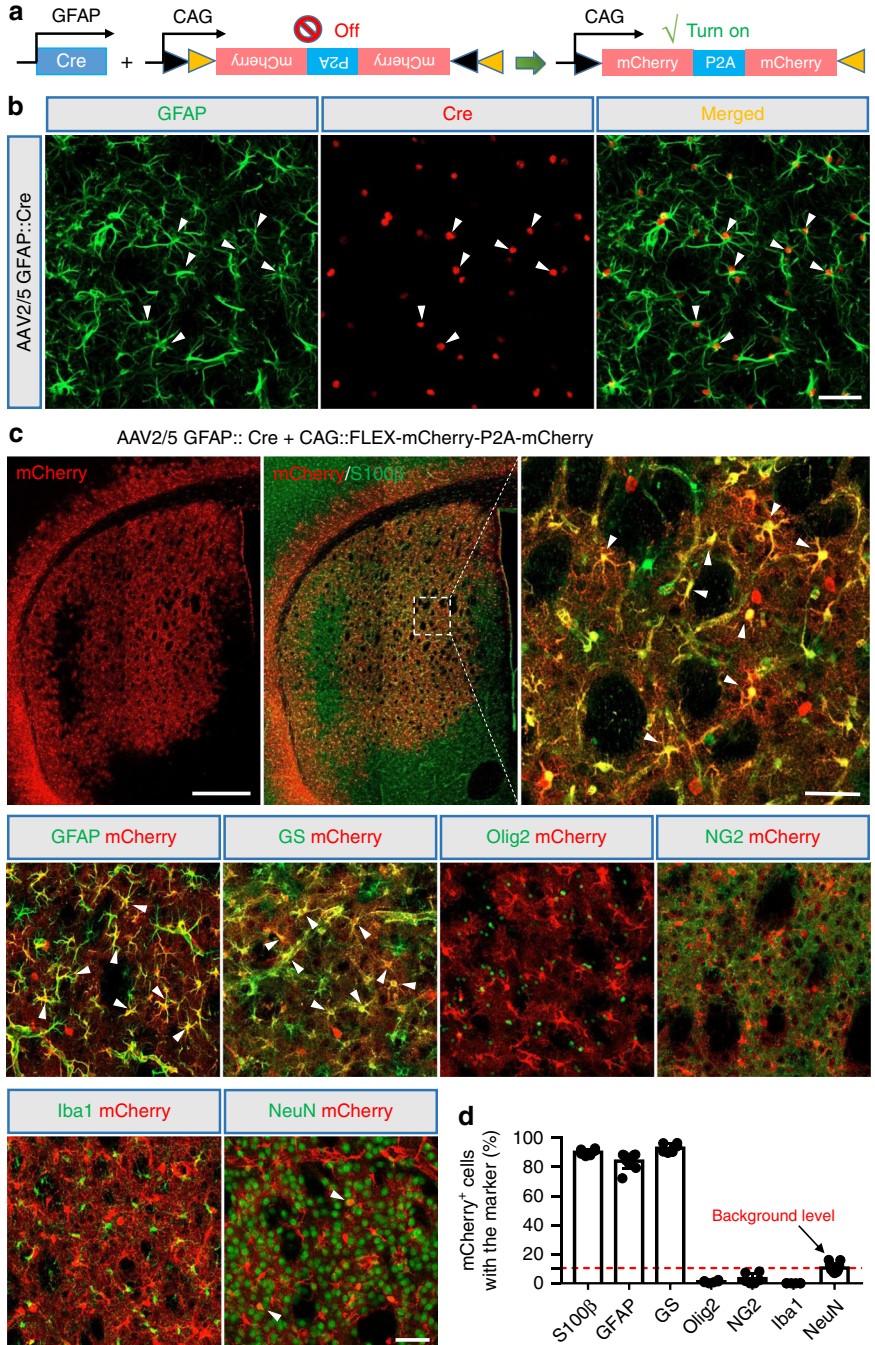

**Fig. 1 Engineered AAV2/5 Cre-FLEx system infects striatal astrocytes specifically in the adult mouse brain. a** Schematic diagram of our engineered AAV2/5 constructs (GFAP::Cre and FLEx-CAG::mCherry-P2A-mCherry) used to target astrocytes specifically with GFAP promoter-controlled expression of Cre recombinase, which in turn will activate the expression of mCherry. **b** Cre recombinase (red) was detected specifically in GFAP+ astrocytes (green) at 7 days post viral injection (dpi) of AAV2/5-GFAP::Cre. White arrowheads indicate astrocytes with Cre expression. Scale bar: 50 μm. **c** Tiled confocal image of the striatum after control AAV mCherry injection (top left) (30 dpi), and the overlaid images of mCherry with a variety of glial markers or neuronal marker (NeuN). S100β, GFAP, and glutamine synthetase (GS) are markers for astrocytes; Olig2 for oligodendrocytes; NG2 for NG2 expressing cells; and Iba1 for microglia. Arrowheads indicate some colocalized cells. Scale bar: 0.5 mm for the top tiled low-magnification images, and 50 μm for the high magnification images. **d** Percentage of mCherry+ cells in colocalization with different cell markers in the striatum. Note that the majority of control mCherry virus-infected cells were astrocytes. Data are shown as mean ± SEM.

and 74.1% at 21 dpi (Fig. 2e, f). Parallel to this trend, more and more mCherry+ cells colocalized with NeuN, while less and less mCherry+ cells colocalized with GFAP from 7 dpi (83.5% GFAP+) to 30 dpi (14.2% GFAP+) (Fig. 2e, f). Conversely, in the control group infected by AAV2/5 mCherry alone, most of the mCherry+ cells were GFAP+ astrocytes, with very few of the

mCherry+ cells co-labeled with NeuN signal across the time points (Fig. 2f, Supplementary Fig. 2). Because NeuroD1 or Dlx2 alone can convert astrocytes into neurons[18,21,35], we further compared their individual effects by injecting the mCherry control, NeuroD1, Dlx2, and NeuroD1 plus Dlx2 into WT mouse striatum. We found that expressing either NeuroD1 or Dlx2 alone

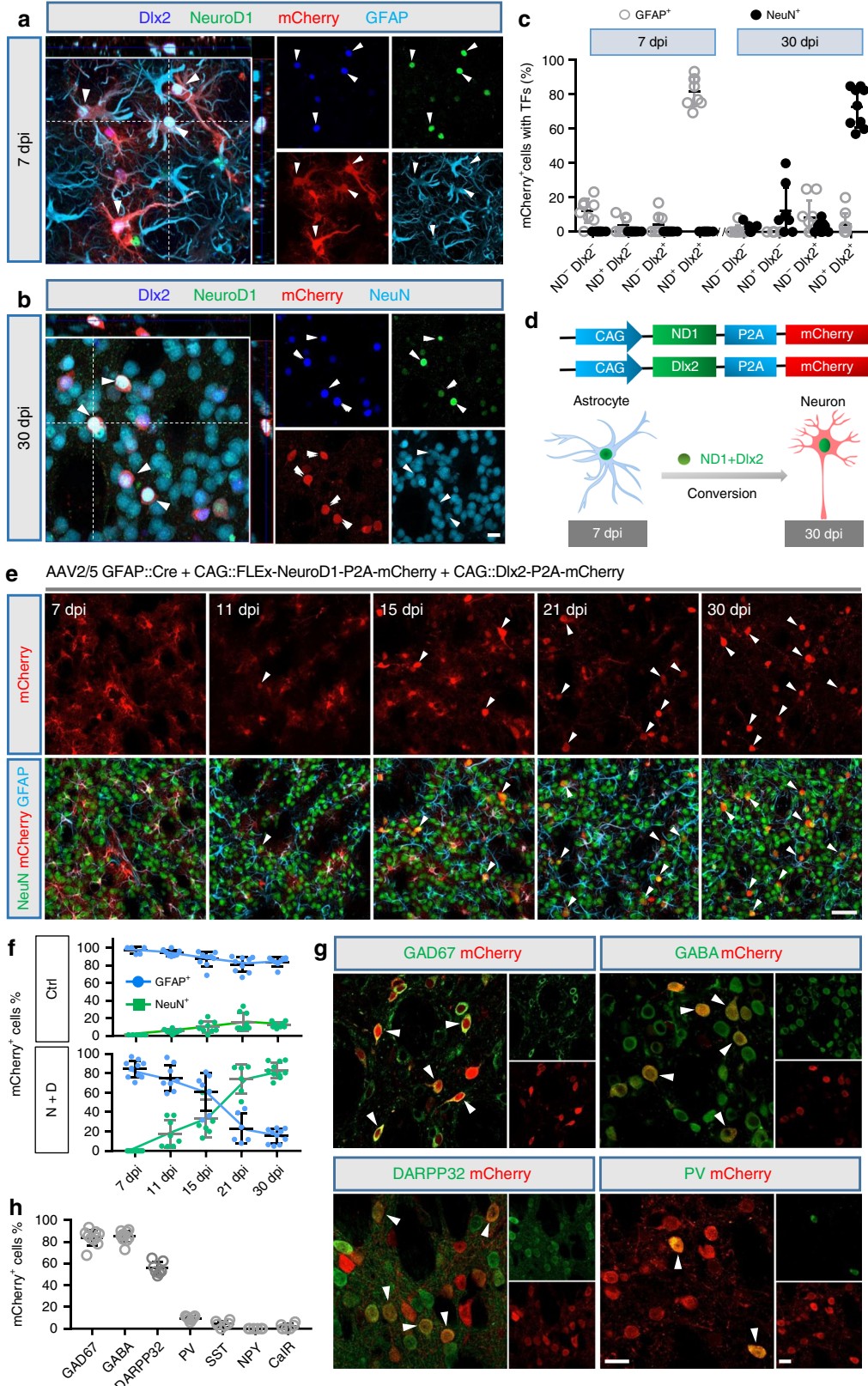

in striatal astrocytes also resulted in a number of the mCherry+ cells co-labeled with NeuN, but the conversion efficiency and the number of converted neurons were much lower than the NeuroD1 + Dlx2 group (Supplementary Fig. 3a–c). These results suggest that NeuroD1 and Dlx2 together have synergistic effects in converting striatal astrocytes into neurons.

To identify the neuronal subtypes after NeuroD1 + Dlx2 induced astrocyte-to-neuron conversion in the striatum, we performed a series of immunostaining experiments with a variety of GABAergic markers including GAD67 and GABA for GABAergic neurons; DARPP32 for MSNs; and paravalbumin (PV), somatostatin (SST), neuronal peptide Y, and calretinin

**Fig. 2 In vivo conversion of striatal astrocytes into GABAergic neurons in WT mouse brain. a** Coexpression of NeuroD1 (green) and Dlx2 (blue) together with mCherry (red, NeuroD1-p2A-mCherry and Dlx2-P2A-mCherry) in AAV-infected striatal astrocytes (GFAP, cyan) at 7 dpi. **b** At 30 dpi, NeuroD1 (green) and Dlx2 (blue) coexpressed cells became NeuN$^+$ neurons (cyan). Scale bar for (**a**) and (**b**): 20 μm. **c** Summarized data showing coexpression of NeuroD1 and Dlx2 in striatal astrocytes at 7 dpi, which mostly converted into NeuN$^+$ neurons by 30 dpi ($n = 8$ mice for 7 dpi, $n = 9$ mice for 30 dpi), data are shown as mean ± SEM. **d** Diagram illustrating the astrocyte-to-neuron conversion process induced by NeuroD1 and Dlx2 coexpression. **e** Representative images illustrating the gradual morphological change from astrocytes to neurons over a time window of 1 month. Note that most mCherry$^+$ cells were co-labeled with GFAP (cyan) at early time points post AAV injection, but later lost GFAP signal and acquired NeuN signal (green). Arrowheads indicate mCherry$^+$ cells that are co-labeled with NeuN. Scale bar: 50 μm. **f** Time course showing the cell identity (astrocyte vs neuron) among viral infected cells (mCherry$^+$ cells) in the control group (mCherry$^+$ alone, top graph, $n = 5$ for 7 dpi, $n = 8$ for 11 dpi, $n = 9$ for 15 dpi, $n = 9$ for 21 dpi, and $n = 8$ for 30 dpi) or NeuroD1 + Dlx2 group (bottom graph, $n = 11$ for 7 dpi, $n = 8$ for 11 dpi, $n = 7$ for 15 dpi, $n = 6$ for 21 dpi, and $n = 10$ for 30 dpi). Most of the viral infected cells in the control group were astrocytes, whereas the NeuroD1 + Dlx2-infected cells gradually shifted from mainly astrocytic population to a mixed population of astrocytes and neurons, and then to mostly neuronal population, data are shown as mean ± SD. **g** Confocal images showing converted neurons co-stained with GAD67, GABA, DARPP32, and parvalbumin (PV) after ectopic expression of NeuroD1 and Dlx2 in striatal astrocytes (30 dpi). Arrowheads indicate co-labeled cells. Scale bar: 20 μm. **h** Quantified data showing the composition of the astrocyte-converted neurons induced by NeuroD1 and Dlx2 in the striatum. Most of the converted neurons were GABAergic neurons (>80%) and a significant proportion were immunopositive for DARPP32 (55.7%), data are shown as mean ± SEM.

(CalR) for striatal interneurons. We found that most of the mCherry$^+$ cells (30 dpi) were GAD67$^+$ (83.9%, $n = 10$ mice) or GABA$^+$ (85.0%, $n = 10$ mice) GABAergic neurons (Fig. 2g, h). Furthermore, the majority of the converted neurons were DARPP32$^+$ (55.7%, $n = 7$ mice; Fig. 2g, h), and a small percentage of the converted neurons were PV$^+$ interneurons (9.6%; Fig. 2g, h), with even fewer other subtypes of interneurons (<5%; Fig. 2h, Supplementary Fig. 4). To conclude, Dlx2 together with NeuroD1 can efficiently convert striatal astrocytes into DARPP32$^+$ GABAergic neurons.

After converting striatal astrocytes into neurons, will the neuron to astrocyte ratio be altered? To answer this question, we analyzed the neuron/astrocyte ratio (Supplementary Fig. 5) and neuron/microglia ratio (Supplementary Fig. 6) in the striatum at 30 days post AAV injection. According to our NeuN and S100β immunostaining, the overall neuronal and astrocytic density as well as the neuron/astrocyte ratio was not significantly changed after astrocyte-to-neuron conversion (Supplementary Fig. 5). One potential explanation for this might be due to the fact that astrocytes are proliferative cells and can divide after neuronal conversion. Indeed, we did observe some S100β$^+$ astrocytes at different stages of cell division in the striatum at 30 days post NeuroD1 + Dlx2 treatment (Supplementary Fig. 5b–d). This might explain why astrocytes did not decrease after neuronal conversion. Similarly, with NeuN and Iba1 immunostaining, we also did not find significant changes in neuronal and microglia density nor the neuron/microglia ratio after astrocyte-to-neuron conversion (Supplementary Fig. 6). Thus, neuronal and glial density are not altered after in vivo cell conversion.

To further validate that the converted neurons originated from astrocytes, we employed GFAP::Cre transgenic mice (Cre77.6, Jackson Lab) to trace cell lineage change by injecting either AAV2/5 FLEx-mCherry alone as a control or AAV2/5 FLEx-NeuroD1-mCherry + FLEx-Dlx2-mCherry for conversion (Fig. 3a, b). As expected, control virus FLEx-mCherry expressed in astrocytes specifically in the Cre77.6 transgenic mouse brain (S100β$^+$, 97.4%, $n = 9$ mice; GFAP$^+$, 94.3%, $n = 8$ mice; GS$^+$, 97.8%, $n = 7$ mice), rather than other types of glial cells or neurons (<5%, $n = 7$ mice for each group; Supplementary Fig. 7a, b). Only <2% of striatal neurons were labeled by mCherry in the control condition ($n = 9$ mice, 3 mice were sacrificed at 28 dpi and 6 mice were sacrificed at 58 dpi). In contrast, injection of NeuroD1 plus Dlx2 viruses into the striatum of Cre77.6 transgenic mice revealed a transitional conversion process at different time points following viral infection. Specifically, mCherry$^+$ cells in ND1 + Dlx2 group showed astrocyte morphology at 7 dpi, with strong GFAP and S100β signal but no NeuN signal (Fig. 3c, d; left column); by

28 dpi, many mCherry$^+$ cells lost GFAP and S100β signal but remained NeuN-negative (GFAP$^-$ & NeuN$^-$ or S100β$^-$ & NeuN$^-$), suggesting a transitional stage (Fig. 3c, d; middle column). At 56 dpi, the majority of mCherry$^+$ cells became NeuN$^+$, suggesting the completion of the astrocyte-to-neuron conversion process (GFAP$^-$ & NeuN$^+$ or S100β$^-$ & NeuN$^+$; Fig. 3c, d, right column). Quantification showed most of the mCherry$^+$ cells were astrocytes (GFAP$^+$ & NeuN$^-$: 97.8%, $n = 6$ mice; S100β$^+$ & NeuN$^-$: 98.1%, $n = 6$ mice) at the beginning (7 dpi), then a number of the transient cells were observed at 28 dpi (GFAP$^+$ & NeuN$^-$: 46.0%, $n = 6$ mice; S100β$^+$ & NeuN$^-$: 47.8%, $n = 6$ mice), and an abundance of mCherry$^+$ neurons were detected at 56 dpi (GFAP$^-$ & NeuN$^+$: 59.1%, $n = 6$ mice; S100β$^-$ & NeuN$^+$: 58.2%, $n = 6$ mice; Fig. 3e, f). Moreover, we also found that most of the ND1 + Dlx2-converted neurons in the striatum of Cre77.6 mice were DARPP32$^+$ MSNs (61.5 ± 2.6%, $n = 8$ mice; Supplementary Fig. 7c). These results in astrocyte-specific Cre transgenic mice further demonstrate that striatal astrocytes can be reprogrammed into MSNs after ectopic expression of NeuroD1 and Dlx2.

**Converting striatal astrocytes into GABAergic neurons in the R6/2 mouse model.** After testing successfully the conversion of striatal astrocytes into GABAergic neurons in the WT mice, we next investigated whether this approach can be used to regenerate GABAergic neurons in a mouse model of HD. In this study, we employed the R6/2 transgenic mouse model for HD, which has been well characterized in terms of the pathogenesis process and widely used for developing therapeutic interventions[7]. To regenerate GABAergic neurons in the striatum of R6/2 mice (both female and male), we injected AAV2/5 NeuroD1 and Dlx2 together in mice at 2 months old when the HD mice started to show neurological phenotypes. One month after viral injection, in the mCherry control group, we observed many infected cells (mCherry$^+$) with astrocyte-like morphology and immunopositive for S100β (Fig. 4a, left panel; and Fig. 4b, top row); while NeuroD1 + Dlx2-infected cells (mCherry$^+$) became immunopositive for NeuN (Fig. 4a, right panel; and Fig. 4b, bottom row). Quantified data showed that in the control group, 86.7% ($n = 6$ mice) of mCherry$^+$ cells were labeled by S100β, and only 9.2% of cells ($n = 6$ mice) were labeled by NeuN (Fig. 4c, cyan bar). In contrast, in NeuroD1 + Dlx2-treated mice, 78.6% ($n = 7$ mice) of viral infected cells were labeled by NeuN while only 15.3% of mCherry$^+$ cells were labeled by S100β (Fig. 4c, green bar). Therefore, these results demonstrate that the striatal astrocytes in the R6/2 mouse brains can be converted into neurons with high efficiency.

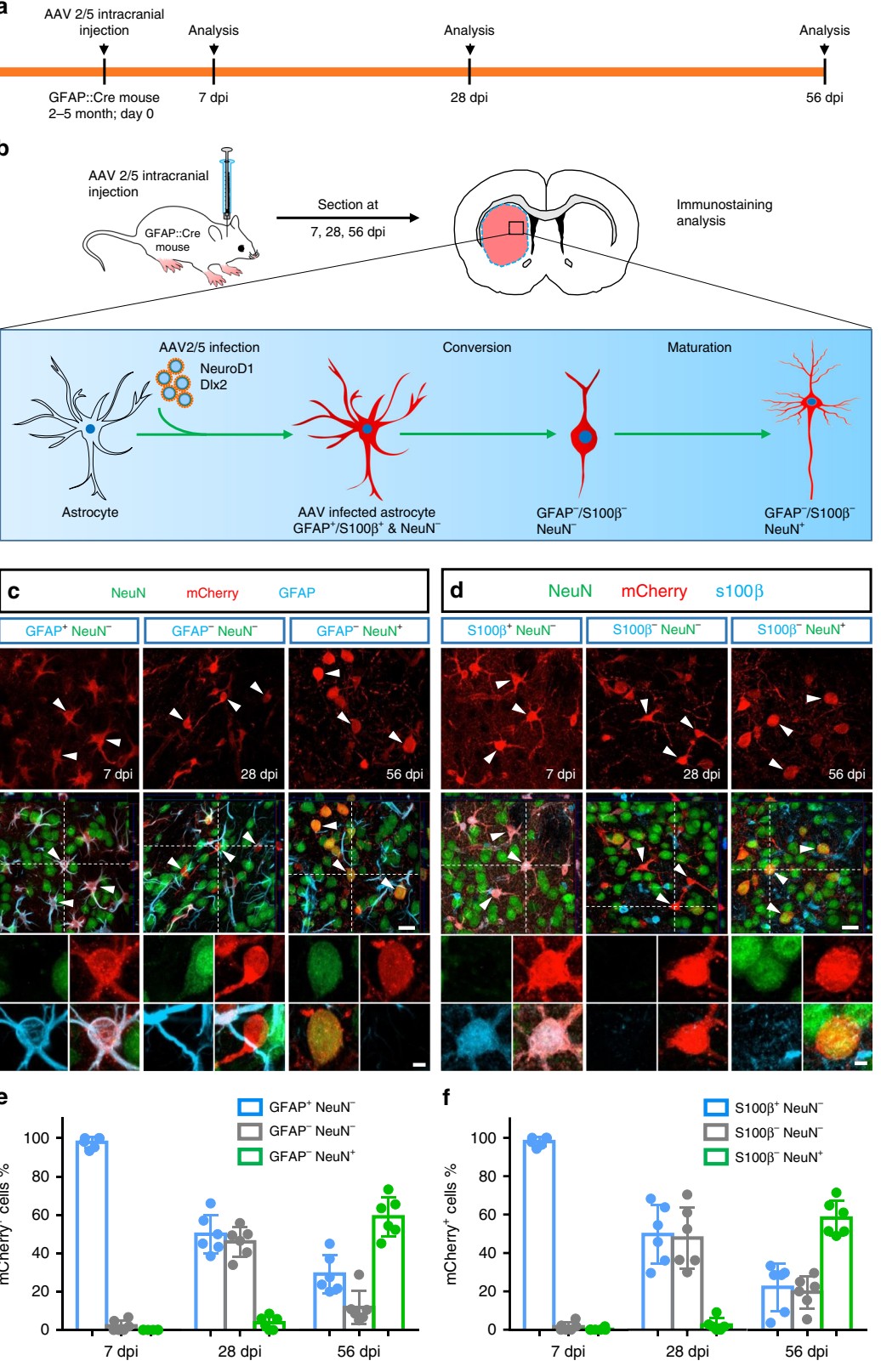

Next, we co-stained mCherry with a variety of GABAergic markers to determine what specific subtypes of GABAergic neurons were converted from astrocytes in R6/2 mouse striatum after injecting NeuroD1 + Dlx2 AAV2/5 (38 dpi). We found that the majority of astrocyte-converted neurons were immunopositive for GAD67 (82.4%, $n = 8$ mice) or GABA (88.7%, $n = 8$ mice;

Fig. 4d–f), suggesting GABAergic neuron identity. Furthermore, 56.6% of the converted cells were DARPP32-positive MSNs ($n = 9$ mice, Fig. 4e, f). There were also a few astrocyte-converted neurons immunopositive for PV (8.4%, $n = 9$ mice; Fig. 4f), but they were rarely positive for SST, NPY, and CalR (all < 5%; Fig. 4f and Supplementary Fig. 8). These results demonstrate that ectopic

**Fig. 3 Converted neurons originate from astrocytes traced by GFAP::Cre 77.6 transgenic mice. a, b** Experimental timeline (**a**) and schematic diagram (**b**) illustrating the use of GFAP::Cre reporter mice to investigate the astrocyte-to-neuron conversion process in the striatum induced by NeuroD1 + Dlx2 (FLEx-NeuroD1-P2A-mCherry and FLEx-Dlx2-P2A-mCherry). **c** Typical confocal images showing the mCherry$^+$ cells (NeuroD1 + Dlx2) co-stained with GFAP and NeuN at 7 dpi (left column), 28 dpi (middle column), and 56 dpi (right column). Scale bar: 20 μm. Insets show a typical cell with different markers. Scale bar: 4 μm. **d** Confocal images of mCherry$^+$ cells (NeuroD1 + Dlx2) co-stained with S100β and NeuN at 7, 28, and 56 dpi. Scale bar: 20 μm. Inset scale bar: 4 μm. **e, f** Quantified data showing a gradual transition from astrocytes to neurons over the time course of 2 months in the GFAP::Cre mice after injection of NeuroD1 and Dlx2 viruses. Note that besides a decrease of astrocytes and an increase of neurons among NeuroD1 and Dlx2-infected cells, about 40% of the infected cells were caught at a transitional stage at 28 dpi, which showed both GFAP-negative and NeuN-negative (**e**, gray bar) or both S100β-negative and NeuN-negative (**f**, gray bar). Also note that the time course of astrocyte-to-neuron conversion is slower in GFAP::Cre mice compared with that induced by GFAP::Cre AAV2/5, both in combination with AAV2/5 FLEx-NeuroD1-P2A-mCherry and FLEx-Dlx2-P2A-mCherry. Data are shown as mean ± SD.

expression of NeuroD1 plus Dlx2 in the striatal astrocytes of R6/2 mice can regenerate a significant number of MSNs for therapeutic treatment.

We further investigated whether in vivo astrocyte-to-conversion could change the glial and neuronal density in the striatum of R6/2 mice. We analyzed the cellular density of neurons and astrocytes as well as neuron/astrocyte ratio (Supplementary Fig. 9) and neuron/microglia ratio (Supplementary Fig. 11) in R6/2 mice with or without cell conversion. Similar to the wild-type mouse striatum, we did not find any significant change in the cellular density nor the neuron/glia ratio in the striatum of R6/2 mice after in vivo cell conversion. We also observed a number of dividing astrocytes in the R6/2 mouse striatum after NeuroD1 + Dlx2 treatment (Supplementary Fig. 9b–d), suggesting that the astrocyte-to-neuron conversion may stimulate proliferation of astrocytes. To test this possibility, we compared the Ki67-labeled dividing astrocytes between control and NeuroD1 + Dlx2 group in R6/2 mouse striatum (30 dpi). We found that compared with the control group, the number of Ki67$^+$ astrocytes in NeuroD1 + Dlx2 group was significantly increased by ~15-fold ($p < 0.001$, unpaired Student's $t$-test; Supplementary Fig. 10). These data suggest that in vivo cell conversion facilitates astrocytic proliferation, explaining why astrocytes are never depleted in the converted areas.

**Converting striatal astrocytes into GABAergic neurons in the YAC128 mouse model for HD.** Since R6/2 mice are a very severe form of HD mouse model that die too soon to mimic the slow progressing disease symptoms in human patients, we further tested the in vivo astrocyte-to-neuron conversion using a more slowly developing HD mouse model YAC128 at an old age of ~15 months (Supplementary Fig. 12). In this study, we injected hGFAP::GFP control virus into the left striatum and hGFAP::NeuroD1 + hGFAP::Dlx2 into the right striatum of the same YAC128 mice (Supplementary Fig. 12a). At 2 months after viral injection, the control GFP-infected cells were mostly GFAP$^+$ astrocytes, whereas the majority of cells infected by NeuroD1 + Dlx2 had lost GFAP signal and half of them became NeuN$^+$ neurons (Supplementary Fig. 12b, c; quantified in Supplementary Fig. 12d). Furthermore, we found that about half of the converted neurons induced by NeuroD1 + Dlx2 were immunopositive for GABA, and 29.8% of the converted neurons were DARPP32$^+$ cells, with only 3.9% of converted neurons being PV$^+$ (Supplementary Fig. 12e, f; $n = 3$). Together, these results demonstrate that the striatal astrocytes in the very old YAC128 mouse brains (15 months old) can still be converted into GABAergic neurons.

**Functional analysis of converted striatal neurons in the R6/2 mouse brain.** We then assessed the functional properties of astrocyte-converted neurons induced by NeuroD1-mCherry and Dlx2-mCherry (mCherry-positive; Fig. 5a) in comparison to the native striatal neurons (neighboring non-converted neurons) in acute striatal slices from R6/2 mice (mCherry-negative; Fig. 5a)

using whole-cell recordings at 30–32 dpi following AAV infection. Striatal neurons from WT mice were also recorded for comparison (Fig. 5 and Supplementary Fig. 13). We first compared the Na$^+$-K$^+$ currents (Fig. 5b–g) and found that there was no significant difference of Na$^+$ currents between converted and neighboring non-converted neurons in R6/2 mice, but both significantly smaller than that recorded in WT mice (Fig. 5f). For K$^+$ currents, converted neurons showed similar amplitude to the WT neurons, while the non-converted neurons of R6/2 mice showed slightly smaller amplitude (Fig. 5g; $n > 15$ neurons from 3 mice per group). Next, for action potential firing, we found that 17 out of 18 mCherry$^+$ cells, and 17 native neurons, were able to fire repetitive action potentials when evoked by step current injection (Fig. 5c, a total of 35 cells from 3 mice were recorded). Regarding basic electrical properties such as the cell membrane input resistance, cell membrane capacitance, resting membrane potential (RMP), action potential (AP) threshold, AP amplitude, and AP frequency, we found no significant differences between native and converted neurons in the R6/2 mice (Fig. 5h–m). When compared with the striatal neurons in the WT mice, the converted neurons in the R6/2 mice had higher input resistance, lower cell capacitance, lower resting membrane potential, and lower action potential amplitude (Fig. 5h–m), suggesting that at 1 month after conversion, these newly converted neurons have not fully matured yet.

Different subtypes of GABAergic neurons have distinct AP firing pattern characteristics[37]. When analyzing the AP firing pattern of the astrocyte-converted neurons, excluding the single mCherry$^+$ cell incapable of firing an AP, most of the converted neurons (72.2%) showed a regular firing frequency (<80 Hz, $n = 13$) with a long delay to the initial AP spike upon stimulation[30,38] (Fig. 5c–r), consistent with a typical MSN firing pattern in the striatum. We also found that 22.2% of converted neurons displayed a fast firing frequency (>80 Hz, $n = 4$; Fig. 5r), consistent with a typical PV neuron firing pattern. Moreover, we further investigated whether astrocyte-converted neurons could be incorporated in local synaptic circuits by examining spontaneous postsynaptic currents (sPSCs), which represent functional synaptic inputs to the converted neurons. As shown by the representative traces (Fig. 5d, e), both spontaneous excitatory postsynaptic currents (sEPSCs) and spontaneous inhibitory postsynaptic currents (sIPSCs) were detected in all native neurons ($n = 9$ from 3 mice) and converted neurons ($n = 11$ from 3 mice). Furthermore, quantitative analyses found no significant differences in the frequency and amplitude of both sEPSCs and sIPSCs among the native neurons and converted neurons in the R6/2 mice (Fig. 5n–q) as well as the striatal neurons in the WT mice (Supplementary Fig. 13c, d). Together, our electrophysiological analyses suggest that striatal astrocytes in the R6/2 mouse brain can be converted into functional GABAergic neurons and integrate into local synaptic circuits.

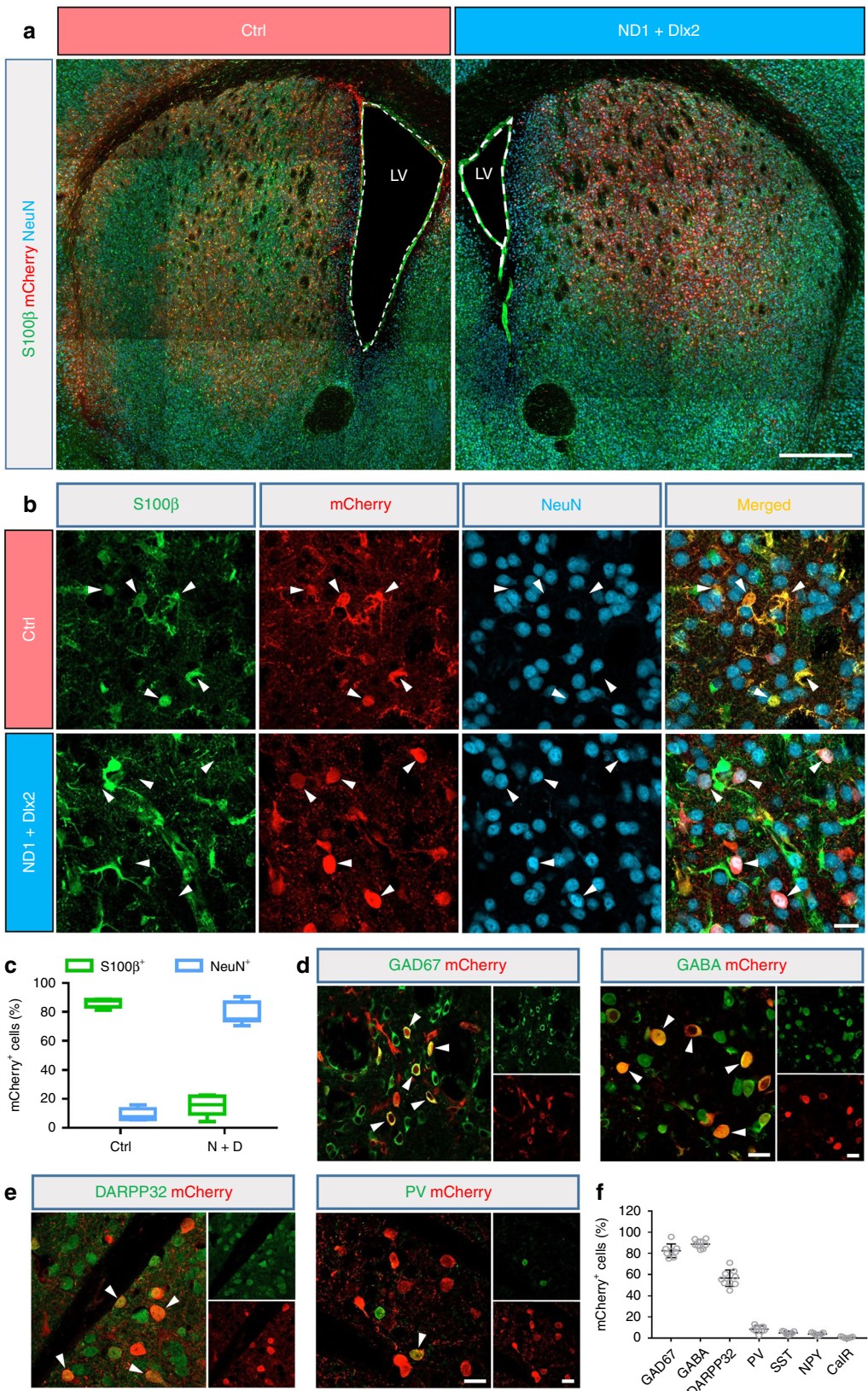

**Axonal projections of the astrocyte-converted neurons**. Striatal MSNs send axonal projections to two distinct nuclei within the basal ganglia, the external globus pallidus (GP) and the substantia nigra pars reticulata (SNr). Due to the severe loss of MSNs in the striatum, these two output pathways are severely disrupted in the HD brain[6,39]. We therefore investigated whether the astrocyte-converted neurons in the striatum could send their axonal projections into these two targets. Indeed, we found a clear mCherry+ axonal tract extending from the striatum to the GP and SNr in NeuroD1 plus Dlx2-treated R6/2 mice (Fig. 6a and Supplementary

**Fig. 4 In vivo conversion of striatal astrocytes into GABAergic neurons in the R6/2 mouse brain. a** The low-magnification coronal sections of the striatum from a pair of littermates of R6/2 mice injected with either control mCherry AAV (left panel) or NeuroD1 plus Dlx2 AAV (right panel) at 30 dpi. Note that the lateral ventricle enlargement was detected clearly in the R6/2 mouse injected with control AAV. Scale bar: 0.5 mm. **b** Higher-magnification images of mCherry$^+$ cells co-stained with S100β (green) and NeuN (cyan). Arrowheads indicate mCherry$^+$ cells co-labeled with S100β in the control group (top row), but in NeuroD1 plus Dlx2 group became co-labeled with NeuN (bottom row). Scale bar: 20 μm. **c** Summary of data showing that by 30 dpi, the majority of mCherry$^+$ cells in the control group were S100β$^+$ astrocytes, while in the NeuroD1 plus Dlx2 group most of the mCherry$^+$ cells were converted into NeuN$^+$ neurons. Data are shown as box plot (boxes, 25-75%; whiskers, 10-90%; lines, median). **d** Most of the striatal astrocyte-converted neurons in the R6/2 mice were immunopositive for GAD67 and GABA. Scale bar: 20 μm. **e** Many of the converted neurons were co-stained by DARPP32 and a few also co-stained with parvalbumin (PV). Scale bar: 20 μm. **f** Quantified data showing that >80% of the converted neurons in the striatum of R6/2 mice were immunopositive for GAD67 and GABA, with a significant proportion also immunopositive for DARPP32 (56.6%) and a smaller percentage being PV$^+$ (8.4%), but very few other GABAergic subtypes.

Fig. 14), but such mCherry$^+$ axonal tract was not detected in the control mCherry alone-injected mice (Supplementary Fig. 15a). Further immunostaining showed that the mCherry$^+$ puncta (axonal nerve terminals) in both the GP and SNr were co-labeled with vGAT, a marker of pre-synaptic GABAergic nerve terminals (Fig. 6b). Quantified data showed that the intensity of the vGAT in the GP and SNr were significantly increased in NeuroD1 plus Dlx2-treated R6/2 mouse brains (Fig. 6c and Supplementary Fig. 15b). These findings demonstrate that the astrocyte-converted neurons can send out GABAergic nerve projections to strengthen GABAergic outputs from the striatum to the GP and SNr in the R6/2 mouse brain.

To further investigate the progress of axonal projections after NeuroD1 + Dlx2 induced in vivo conversion in the R6/2 mouse brain, we injected a retrograde tracer, cholera toxin subunit B (CTB), into the GP or SNr at two different time points, 21 or 30 dpi. At 7 days post CTB injection, the mice were sacrificed for analysis of the CTB-labeled neurons in the striatum (see schematic illustration in Fig. 6d). Sagittal brain sections were made for validating the CTB injected sites (Supplementary Fig. 16). When CTB was injected at 21 dpi, we found a number of CTB-labeled native neurons (NeuN$^+$, mCherry$^-$) in the striatum as expected, but very few of the converted neurons (NeuN$^+$, mCherry$^+$) were labeled by CTB (GP = 8.2%, $n$ = 509 from 5 mice; SNr = 3.5%, $n$ = 483 from 5 mice; Fig. 6e–g). However, when CTB was injected at 30 dpi, we found that CTB was not only detected in native neurons but also in converted neurons (Fig. 6e, f). Quantified data showed that the percentage of CTB-labeled converted neurons was significantly increased when CTB was injected at 30 dpi compared with 21 dpi (GP = 27.7%, $n$ = 535 neurons from 5 mice, $p$ = 0.014; SNr = 29.4%, $n$ = 511 neurons from 5 mice, $p$ = 0.004, unpaired Student's $t$-test; Fig. 6g). Therefore, these data demonstrate that the in vivo converted MSNs can extend their axonal projections into the GP and SNr in the R6/2 mouse brain.

**Alleviation of neurodegeneration in R6/2 mice by in vivo cell conversion.** Huntington's disease is an autosomal dominant disorder associated with a mutation in the gene encoding huntingtin (Htt). The mutation leads to excessive polyglutamine repeats yielding mutant Htt (mHtt) aggregation and subsequent neurodegeneration, particularly in the striatum. Because the newly generated neurons are converted from astrocytes and mHtt aggregation has been detected both in neurons and astrocytes in R6/2 mouse striatum[40], we compared the progress of mHtt inclusions in striatal astrocytes and neurons at age 60 days (P60) and 90 days (P90) in the R6/2 mouse striatum. We found that mHtt nuclear inclusions were detected at P60 in 20.6% of S100β positive astrocytes and 71.1% in neurons (Supplementary Fig. 17a, b). At 3 months old, 35.8% astrocytes and 75.5% of neurons displayed mHtt inclusions (Supplementary Fig. 17a, b). These data suggest that astrocytes have less mHtt inclusions than

neurons in the R6/2 mouse striatum. Interestingly, we found that the astrocyte-converted neurons (51.1%, $n$ = 151 neurons from 12 mice) displayed less mHtt inclusions when compared with the native neurons (77.1%, $n$ = 655 neurons from 12 mice; $p$ < 0.002, one-way ANOVA with Bonferroni's post hoc test), or the neurons in the control group (80.3%, $n$ = 709 neurons from 11 mice; $p$ < 0.001, one-way ANOVA with Bonferroni's post hoc test) (Fig. 7a,c). These data indicate that in the R6/2 mouse striatum, neurons have more mHtt nuclear inclusions than astrocytes and the astrocyte-converted neurons have less mHtt nuclear inclusions than preexisting neurons.

Striatum atrophy caused by neurodegeneration has been reported previously in the R6/2 mouse brain[41]. Consistently, we also observed striatal atrophy in R6/2 mice compared with their WT littermates (Supplementary Fig. 17c). Quantification data showed a 31.8% reduction in the striatum volume in 3-month-old R6/2 mice ($n$ = 9 mice, $p$ < 0.001, one-way ANOVA with Bonferroni's post hoc test; Fig. 7d). Remarkably, we found that the striatum atrophy was alleviated in the NeuroD1 plus Dlx2-treated R6/2 mice compared with the control virus-treated R6/2 mice (Fig. 7b; AAV2/5 were injected at P60, mice were sacrificed at P98). Quantified data showed 30.3% striatum atrophy in the control virus-treated group ($n$ = 6 mice), but only 16.9% striatum atrophy in the NeuroD1 plus Dlx2 group ($n$ = 7 mice, $p$ = 0.004, one-way ANOVA with Bonferroni's post hoc test; Fig. 7d). Therefore, these results suggest that the in vivo astrocyte-to-neuron conversion therapy can reduce the striatum atrophy in R6/2 mice.

**Attenuation of phenotypic deficits in R6/2 mice by in vivo cell conversion.** The R6/2 mice display a progressive neurological phenotype that mimics many of the features of HD patients[7]. We therefore examined whether the in vivo cell conversion approach could alleviate the abnormal phenotypes in the R6/2 mice using a series of behavioral tests. We first performed the catwalk behavioral test to evaluate the gait changes in the R6/2 mice in comparison to their WT littermates (P90–97). We found that the average stride length was significantly reduced in the R6/2 mice when compared with WT littermates (WT = 5.80 ± 0.30 cm, $n$ = 13 mice, 6 male and 7 female; R6/2 = 3.91 ± 0.11 cm, $n$ = 10, 3 male and 7 females; $p$ < 0.001, one-way ANOVA with Bonferroni's post hoc test; Fig. 8a, b and Supplementary movie 1–2). To test the effect of our gene therapy, R6/2 mice received intracranial AAV2/5 injection bilaterally at P60 and after 30–37 days post viral injection underwent the catwalk behavioral test (Fig. 8k). We found that the stride length was significantly improved in the NeuroD1 plus Dlx2-treated mice (4.91 ± 0.13 cm, $n$ = 19, 8 males and 11 females; $p$ < 0.001, one-way ANOVA with Bonferroni's post hoc test), compared with the control AAV2/5 mCherry-injected mice (3.95 ± 0.14 cm, $n$ = 13, 6 males and 7 females; Fig. 8a, b and Supplementary movie 3–4). There was no significant difference in footprint width between different groups (Fig. 8a,c). We also assessed the locomotion activity

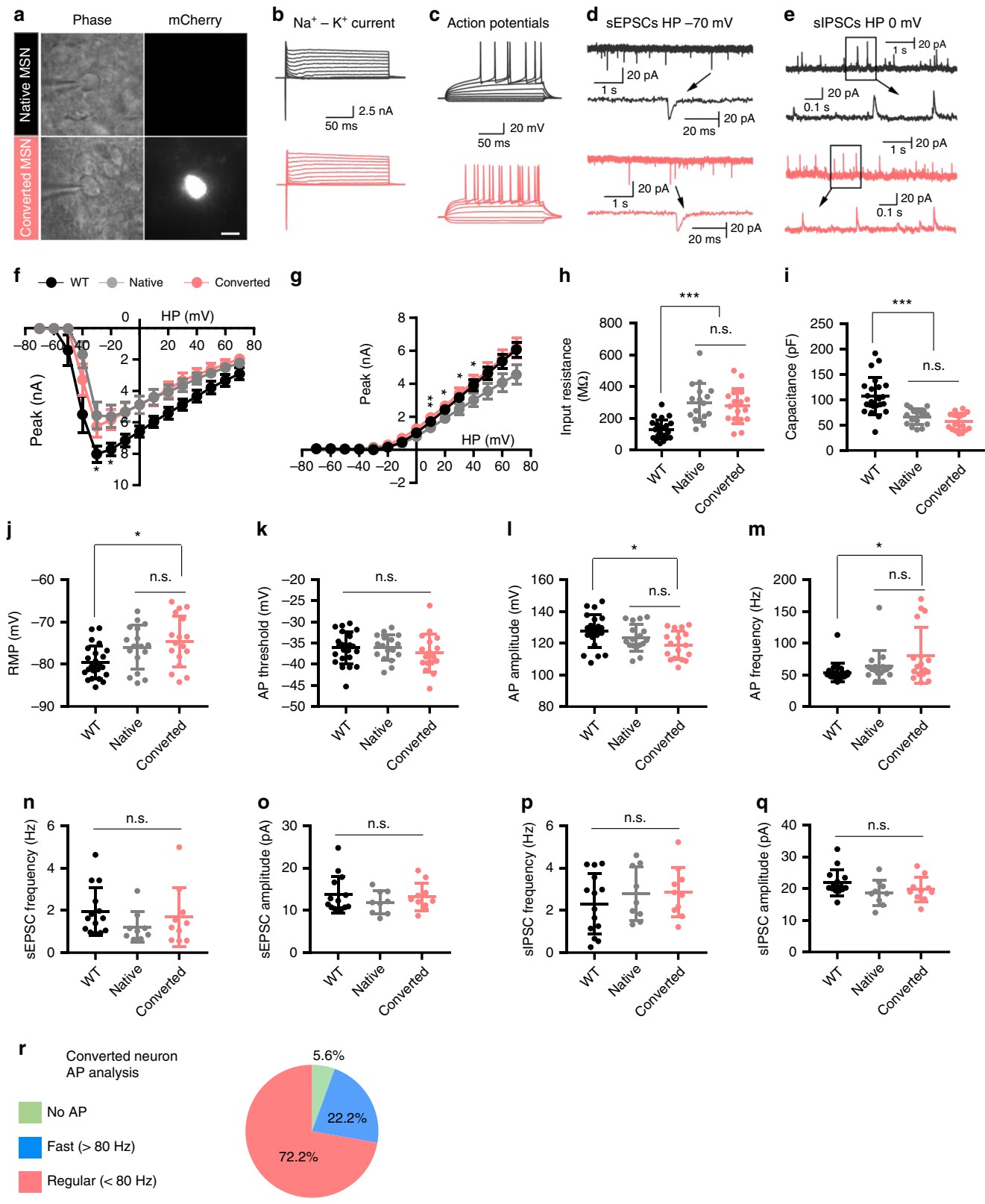

with the open field test. We found that the total travel distance of R6/2 mice (in 20 min) showed a dramatic decrease (1886 ± 252 cm, $n = 12$, 5 males and 7 females; $p < 0.001$, one-way ANOVA with Bonferroni's post hoc test) compared with the WT littermates (6163.8 ± 263.0 cm, n = 14, 7 males and 7 females; Fig. 8d, e).

Remarkably, the walking distance showed a significant increase in the NeuroD1 plus Dlx2-treated R6/2 mice (3648 ± 367 cm, $n = 18$ mice, 10 male and 8 female mice), compared with the mCherry-treated R6/2 mice (2023 ± 331 cm, $n = 12$ mice, 5 male and 7 female mice; one-way ANOVA with Bonferroni's post hoc test;

**Fig. 5 Functional characterization of the striatal astrocyte-converted neurons in the R6/2 mouse brain slices. a** Phase and fluorescent images of a native neuron (mCherry$^-$, top row) and a converted neuron (mCherry$^+$, bottom row). Scale bar: 10 μm. **b** Representative traces of Na$^+$-K$^+$ currents recorded in native (gray) and converted neurons (red). **c** Repetitive action potentials (AP) evoked by step-wise current injections. Note a significant delay to the initial action potential firing upon depolarization stimulation in both native and converted neurons. Such delayed firing is a typical MSN electrophysiological property. **d, e** Typical traces of sEPSCs and sIPSCs recorded from native (gray traces, top row) and converted neurons (red traces, bottom row). **f, g** I–V plot of Na$^+$-K$^+$ currents recorded from striatal neurons in the viral-injected R6/2 mice and non-treated WT mice. The Na$^+$ currents in both converted (red) and non-converted striatal neurons (gray) in the R6/2 mice were smaller than that recorded from the striatal neurons in the WT mice (black). The K$^+$ current in converted neurons was significantly larger than that in non-converted neurons in the R6/2 mouse striatum (unpaired Student's $t$-test). $*p < 0.05$, $**p < 0.01$. Data are shown as mean ± SEM. **h–m** Summary graphs in scatter-plot showing electrical properties among the converted (red dots) and non-converted neurons (gray dots) in the R6/2 mice, together with the wild-type neurons (black dots): input resistance (**h**), capacitance (**i**), resting membrane potential (**j**), AP threshold (**k**), AP amplitude (**l**), and AP frequency (**m**). There were no significantly differences between the converted and non-converted neurons in the R6/2 mice, but neurons from R6/2 mice showed some differences from the wild-type neurons. One-way ANOVA with Bonferroni's post hoc test. **n–q** Summary graphs in scatter-plot showing similar synaptic responses among the wild-type neurons (black dots), and the converted (red dots) and non-converted neurons (gray dots) in the R6/2 mice: sEPSC frequency (**n**), sEPSC amplitude (**o**), sIPSC frequency (**p**), and sIPSC (**q**). $p > 0.4$ for all groups, one-way ANOVA with Bonferroni's post hoc test. **r** Pie chart showing the percentage of neurons with different firing pattern among the converted neurons.

Fig. 8d, e). These data suggest that the in vivo cell conversion approach significantly improves the motor functions of the R6/2 mice.

In addition, we also examined the body weight, clasping behavior, and grip strength of the R6/2 mice after gene therapy treatment. R6/2 mice have been reported to lose body weight at 8 weeks old[42]. To test our gene therapy effects, the R6/2 mice were randomly divided into two groups, and the body weight was measured 7 days before surgery. No significant difference was found between the two groups ($p = 0.367$; Fig. 8f). Interestingly, the R6/2 mice that were treated with NeuroD1 plus Dlx2 lost less body weight than the R6/2 mice that were injected with the control virus at 30 dpi (Ctrl = $21.13 \pm 0.39$ g, $n = 25$, 9 females and 16 males; N + D = $22.42 \pm 0.38$ g, $n = 28$, 11 females and 17 males; $p = 0.021$, unpaired Student's $t$-test; Fig. 8f). Next, we used the paw clasp test to measure dystonia and dyskinesia in the R6/2 mice. The typical clasping phenotype (Fig. 8g, top panel) was observed in most of the R6/2 mice. However, the percentage of R6/2 mice showing clasping was significantly reduced after NeuroD1 plus Dlx2 treatment (Ctrl = 88.2%, $n = 34$, 14 females and 20 males; N + D = 67.7%, $n = 31$, 13 females and 18 males; $p = 0.045$, 2-sided Pearson Chi-square test; Fig. 8h). Moreover, the clasping score was also significantly decreased in NeuroD1 plus Dlx2 group (Ctrl = $3.4 \pm 0.4$, $n = 34$, 14 females and 20 males; N + D = $2.3 \pm 0.4$, $n = 31$, 13 females and 18 males; $p = 0.040$, unpaired Student's $t$-test; Fig. 8i and Supplementary movie 5–6). We further measured the grip strength and found that there was no significant difference between the control virus-treated and the NeuroD1 plus Dlx2-treated R6/2 mice (Fig. 8j). Remarkably, when the survival rate of R6/2 mice was analyzed at 38 dpi (viral injection at 2 months old), 93.9% of the R6/2 mice that were injected with NeuroD1 plus Dlx2 were still alive, but only 55.2% of the R6/2 mice that received control AAV2/5 mCherry injection were alive, which is expected for R6/2 mice at this age ($P < 0.001$, 2-sided Pearson Chi-square test; Fig. 8l). Altogether, these results demonstrate that in vivo regeneration of GABAergic neurons in the striatum of R6/2 mice can partially rescue the phenotypic deficits and extend the life span.

## Discussion

In this study, we demonstrate that striatal astrocytes in the R6/2 and YAC128 mouse models for HD can be converted into GABAergic neurons and alleviate motor functional deficits. The in vivo converted neurons show similar electrophysiological properties to their neighboring preexisting neurons, and send long-range axonal projections to the GP and SNr in a time-dependent manner. Our NeuroD1 and Dlx2-based gene therapy partially rescued motor

functional deficits and increased the survival of R6/2 mice. This proof-of-principle study illustrates a neuroregenerative gene therapy to treat neurodegenerative disorders through in vivo astrocyte-to-neuron conversion approach.

This work demonstrated the in vivo regeneration of GABAergic neurons, and in particular the DARPP32$^+$ neurons with similar electrophysiological properties to the preexisting neurons. An immediate concern was whether these newly converted neurons were actually preexisting neurons, because AAV itself can infect both astrocytes and neurons. This has been carefully considered but found incompatible with the results we observed. First, consistent with the use of GFAP::Cre to restrict viral expression in astrocytes, we found that essentially all the NeuroD1 + Dlx2-infected cells were indeed GFAP$^+$ astrocytes at early time points (before 7 dpi). These viral infected astrocytes gradually lost GFAP signal while acquired NeuN signal, showing a clear time-dependent transition from astrocytic population to neuronal population. If it is not because of AtN conversion, it is impossible to explain such gradual transition from astrocytes to neurons. Second, our axonal projection experiments also showed a time-dependent retrograde labeling of the newly converted neurons, consistent with gradual maturation and axonal projections from newly generated neurons instead of preexisting neurons. The control AAV mCherry group did not show such time-dependent axonal projections, consistent with the fact that no new neurons were converted in the control group. Together, these results suggest that there is no alternative explanation that can account for the gradual transition from astrocytic population to neuronal population other than direct AtN conversion.

Previous studies have reported that transplanting external cells might generate new neurons in animal models for HD[43]. For example, transplantation of hESC-derived DARPP32$^+$ GABAergic neurons or human NSCs may improved motor functions[10,44]. Besides stem cells, human fibroblast cells may be converted into striatal GABAergic neurons using microRNAs (miR-9/9∗ and miR-124) combined with transcription factors (Ctip2, Dlx1, Dlx2, and Myt1L) before transplanted into the mouse brain[30]. On the other hand, cell transplantation also faces a significant challenge such as long-term survival in a completely different environment from that before transplantation[45]. Besides generating new neurons, whether right neurons are generated in the right place and then project to the right targets to form right circuits is even more critical for brain repair. In comparison to transplantation of external cells, the astrocyte-converted neurons are in the right place from the very beginning. Therefore, the in situ astrocyte-converted neurons may integrate into the local environment much more easily than externally transplanted cells. It is now also

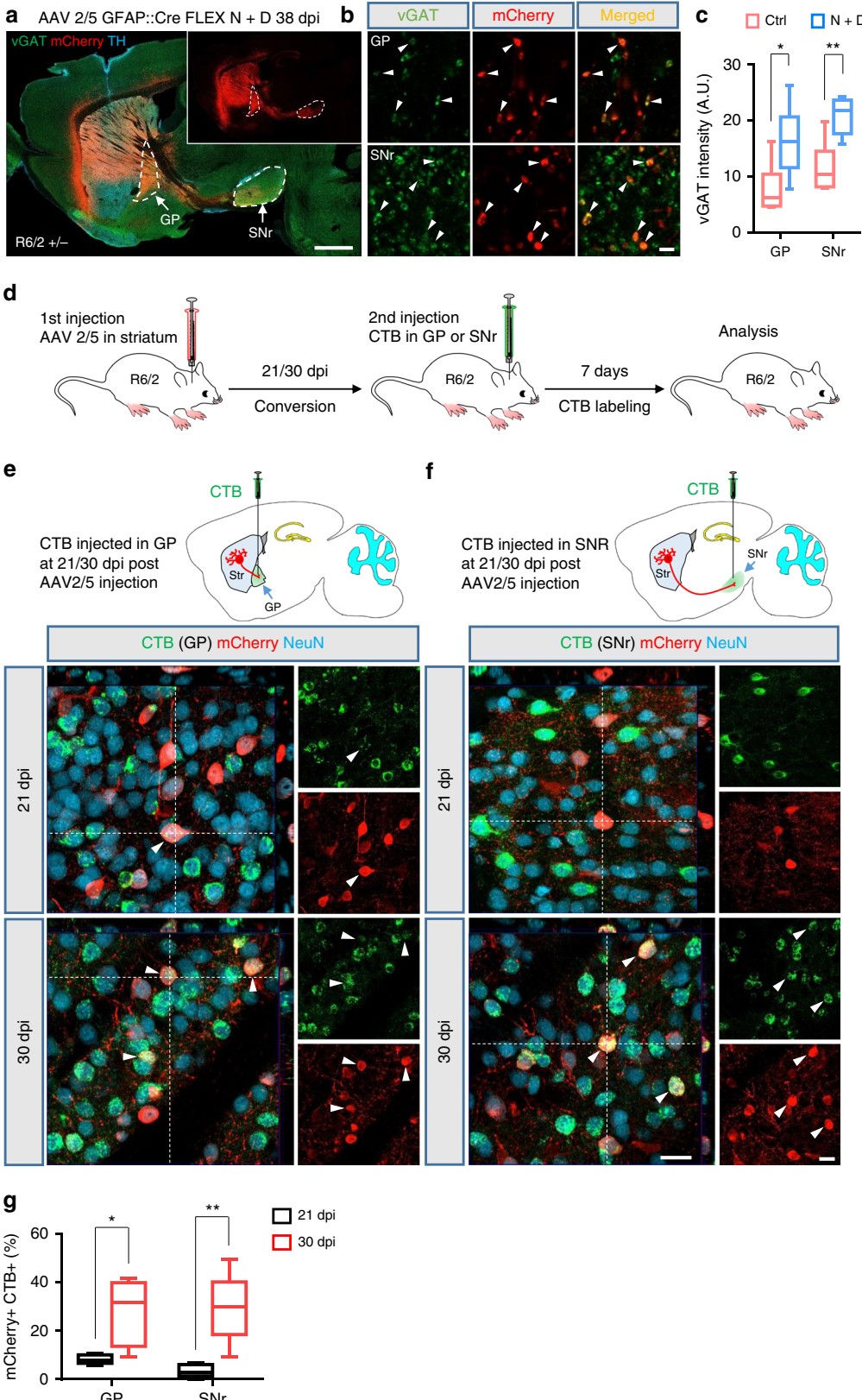

possible to generate a variety of subtypes of neurons with the right combination of transcription factors[46–48]. We demonstrate in this study that combining NeuroD1 and Dlx2 together can generate DARPP32$^+$ neurons in the striatum. The importance of Dlx2 in generating GABAergic neurons has been reported

before[33,34,36], such as the direct conversion of human fibroblast cells into striatal neurons using microRNAs plus transcription factors Dlx1 and Dlx2[30]. Moreover, because astrocytes are widely distributed throughout the CNS and have intrinsic proliferation capability, our in vivo cell conversion technology may offer an

**Fig. 6 Axonal projections of the striatal astrocyte-converted neurons in the R6/2 mouse brain. a** A sagittal view of a R6/2 mouse brain section immunostained for vGAT (green) and tyrosine hydroxylase (TH, cyan). TH positive cell bodies were present in the substantia nigra (above the SNr) and dense TH innervation was observed in the striatum. Inset shows the mCherry channel only to illustrate the axonal projections from the striatum to the GP and SNr. Scale bar: 1 mm. **b** High-resolution images showing mCherry$^+$ puncta co-stained with vGAT (arrowhead) in GP and SNr (38 dpi). Scale bar: 2 μm. **c** Quantified data showing vGAT intensity in the GP and SNr significantly enhanced in NeuroD1 plus Dlx2-treated R6/2 mouse brains. **d** Experimental design of CTB retrograde tracing of converted neurons in the R6/2 mouse brain. Mice were sacrificed for immunohistochemistry analysis at 7 days after CTB injection. **e** Retrograde tracing of striatal astrocyte-converted neurons by injecting CTB into the GP at 21 or 30 days after AAV2/5 NeuroD1 + Dlx2 injection. Few CTB (green)-labeled converted neurons (red) were detected in the striatum at 21 dpi group (arrowhead), but many more CTB-labeled converted neurons were observed at 30 dpi group (arrowheads). **f** CTB injection into the SNr to trace striatal astrocyte-converted neurons. Even fewer converted neurons were labeled by CTB at 21 dpi group, but CTB labeling was clearly identified among the converted neurons in the striatum at 30 dpi group (arrowheads). Note that, in both GP (**e**) and SNr (**f**), many non-converted preexisting neurons were retrograde labeled by CTB, as expected. Scale bar for (**e**) and (**f**): 20 μm. **g** Bar graphs showing the percentage of CTB-labeled converted neurons in the R6/2 mouse striatum, which showed a significant increase from 21 dpi (black bars, immature neurons) to 30 dpi (red bars, more mature neurons). *$p < 0.05$, **$p < 0.01$, unpaired Student's $t$-test. Data are shown as box plot (boxes, 25–75%; whiskers, 10–90%; lines, median).

---

economical way to regenerate a large number of new neurons for the treatment of neurodegenerative disorders such as HD.

Like any new technology, there are of course limitations of the emerging in vivo cell conversion technology. One obvious limitation is that there must be glial cells present for conversion. When injury or degeneration is so severe that a cavity has formed, or when glial cells themselves are significantly damaged, there will be a deficit of glial source to enact the conversion. In such severe injury condition, it may be necessary to transplant external cells to fill the cavity first for tissue repair[49]. Another limitation is for the treatment of diseases caused by gene mutations, such as the *Htt* mutation in R6/2 mouse model for HD used in this study. Converting astrocytes into neurons does not directly address the gene mutation problem. The converted neurons sooner or later may develop the mHtt inclusions and degenerate again. One possible solution is to combine the cell conversion approach together with CRISPR gene editing techniques to correct the gene mutation[15] so that the converted neurons may survive better.

In conclusion, this study demonstrates not only at the cellular level of regenerating GABAergic neurons but also at the functional level of circuit reconstruction and motor behavioral improvement. The significant extension of life span in the R6/2 HD mouse model after our NeuroD1 + Dlx2 gene therapy treatment suggests that a potential disease-modifying therapy is now on the horizon for HD and other neurodegenerative disorders.

## Methods

**Animals.** Animals were housed in a 12:12 h light:dark cycle with free access to chow and water. The R6/2 strain (B6CBA-Tg(HDexon1)62Gpb/3J) was maintained by ovarian transplant hemizygote females × B6CBAF1/J males, both were purchased from Jackson Laboratory. Mice were genotyped by PCR after weaning (P21–27) and the littermates without mutation were used as normal mice (2–5 months). Some of the R6/2 transgenic mice were directly purchased from the Jackson Laboratory at ages of 4–6 weeks. The GFAP::Cre transgenic mice (B6.Cg-Tg(Gfap-cre)77.6Mvs/2J, Cre77.6) were purchased from Jackson Laboratory as well. The 2–5-month-old hemizygous mice were used for AAV injection. Both male and female mice were used in this study. Experimental protocols were approved by the Pennsylvania State University IACUC and in accordance with guidelines of National Institutes of Health.

YAC128 transgenic mice were purchased from Nanjing Biomedical Research Institute of Nanjing University (NBRI). The mice were housed in a 12 h light/dark cycle and supplied with enough food and water. Experimental protocols were approved by Jinan University IACUC.

**AAV production.** Recombinant AAV2/5 was produced in 293 AAV cells (Cell Biolabs). Briefly, polyethylenimine (PEI, linear, MW 25,000) was used for transfection of triple plasmids: the pAAV expression vector, pAAV5-RC (Cell Biolab) and pHelper (Cell Biolab). At 72 h post transfection, cells were harvested and centrifuged. The cells were then cyclically frozen and thawed four times by placing it on dry ice/ethanol and a 37 °C water bath. AAV crude lysate was purified by centrifugation at 54,000 rpm for 1 h in discontinuous iodixanol gradients with a Beckman SW55Ti rotor. The virus-containing layer was extracted and concentrated by Millipore Amicon Ultra Centrifugal Filters. The AAV2/5 genome copies (GC) per

injection for GFAP::Cre is $3.55 \times 10^7$ GC; CAG::FLEx-mCherry-P2A-mCherry is $2.54 \times 10^9$ GC; CAG::FLEx-NeuroD1-P2A-mCherry is $1.59 \times 10^9$ GC and CAG::FLEx-Dlx2-P2A-mCherry is $2.42 \times 10^9$ GC. Virus titer was $7.7 \times 10^{10}$ GC/ml for GFAP::Cre; $1.65 \times 10^{12}$ GC/ml for FLEx-mCherry-P2A-mCherry, $2.07 \times 10^{12}$ GC/ml for FLEx-NeuroD1-P2A-mCherry and $3.14 \times 10^{12}$ GC/ml for FLEx-Dlx2-P2A-mCherry, determined by QuickTiter™ AAV Quantitation Kit (Cell Biolabs).

For AAV injection in YAC128 mice, human GFAP promoter was used to drive GFP and NeuroD1 or Dlx2 expression. hGFAP::GFP and hGFAP::NeuroD1-P2A-GFP were packaged with capsids of serotype 9. hGFAP::Dlx2-P2A-GFP was packaged with capsids of serotype 5. Virus stocks were adjusted to $2–3 \times 10^{12}$ GC/mL in 0.001% F-68.

**Stereotaxic viral injection.** Brain surgeries were conducted on 2–5-month-old wild-type mice or 2-month-old R6/2 mice for AAV injection. The mice were anesthetized by injecting ketamine/xylazine (120 mg/kg and 16 mg/kg) into the peritoneum, followed by fur trimming, and placement into a stereotaxic setup. Artificial eye ointment was applied to cover the eye for protection purposes. Oxygen was provided for the R6/2 mice throughout surgery. The operation began with a midline scalp incision followed by the creation of a (~1 mm) drill hole on the skull for intracranial injection into the striatum (AP +0.6 mm, ML ±1.8 mm, DV −3.5 mm). Each mouse received a bilateral injection of AAV2/5 using a 5 μl syringe and a 34 G needle. The injection volume was 2 μl and the flow rate was controlled at 0.2 μl/min. Some R6/2 mice received secondary surgery after AAV2/5 injection where CTB (ThermoFisher, C34775) was delivered. The mice were anesthetized by 2.5% Avertin (250–325 mg/kg) and oxygen was supplied during the surgery. CTB (0.5 μg/site) was injected into the globus pallidus (AP −0.2 mm, ML 1.8 mm, DV −4.0 mm) or substantia nigra pars reticulata (AP −3.0 mm, ML 1.7 mm, DV −4.0 mm), two target areas of the striatal MSN's projections. After viral injection, the needle was kept in place for at least 10 min before being slowly withdrawn. Coordinates are measured from bregma.

For viral injection in YAC128 mice, 15-month-old YAC128 mice were used for experiments because of its slow disease progression. The mice were anesthetized with 20 mg/kg 2.5% Avertin (a mixture of 25 mg/ml of Tribromoethethanol and 25 μg/ml T-amylalcohol) through intraperitoneal injection and then placed in the stereotaxic frame. Viruses were injected into the striatum (coordinate of AP: +1.0 mm, ML: ±2.0 mm, DV: −3.0 mm), with one side injecting control AAV (hGFAP::GFP) and the other side injected with NeuroD1 + Dlx2. The injection volume and flow rate were controlled at 3 μl at 0.2 μl/min. After injection, the injection pipette was kept in place for about 10 min and then slowly withdrawn.

**Immunohistochemistry and analysis.** For brain slice immunostaining, the animals were deeply anesthetized with 2.5% Avertin and then quickly perfused with ice-cold artificial cerebrospinal fluid (aCSF) to wash away the blood. Then brains were quickly removed and post-fixed in 4% PFA overnight at 4 °C in darkness. After fixation, the samples were cut into 40-μm sections by a vibratome (Leica, VTS1000). Brain slices were washed three times in phosphate buffer solution (PBS, pH: 7.35, OSM: 300) for 10 min each. Blocking was performed for 2 h in 0.3% triton PBS + 5% normal donkey serum (NDS). Primary antibody was diluted in 0.05% triton PBS + 5% NDS and incubated in a moist environment at 4 °C for two nights (see Table 1 for the primary antibody information). After washing three times in PBS, the samples were incubated with appropriate secondary antibodies conjugated to Alexa Flour 405, or Alexa Flour 488, or Cy3, or Alexa Flour 647 (1:500, Jackson ImmunoResearch) for 2 h at room temperature, followed by extensive washing in PBS. The secondary antibody was diluted in 0.05% triton PBS + 5% NDS. For GAD67 and GABA immunostaining, the samples were fixed in 4% PFA and 0.2% glutaraldehyde, the sections were mildly permeabilized in 0.05% Triton PBS for 30 min, and Triton was removed for rest of the immunostaining procedure. The samples were mounted on glass slides and stored at 4 °C in darkness.

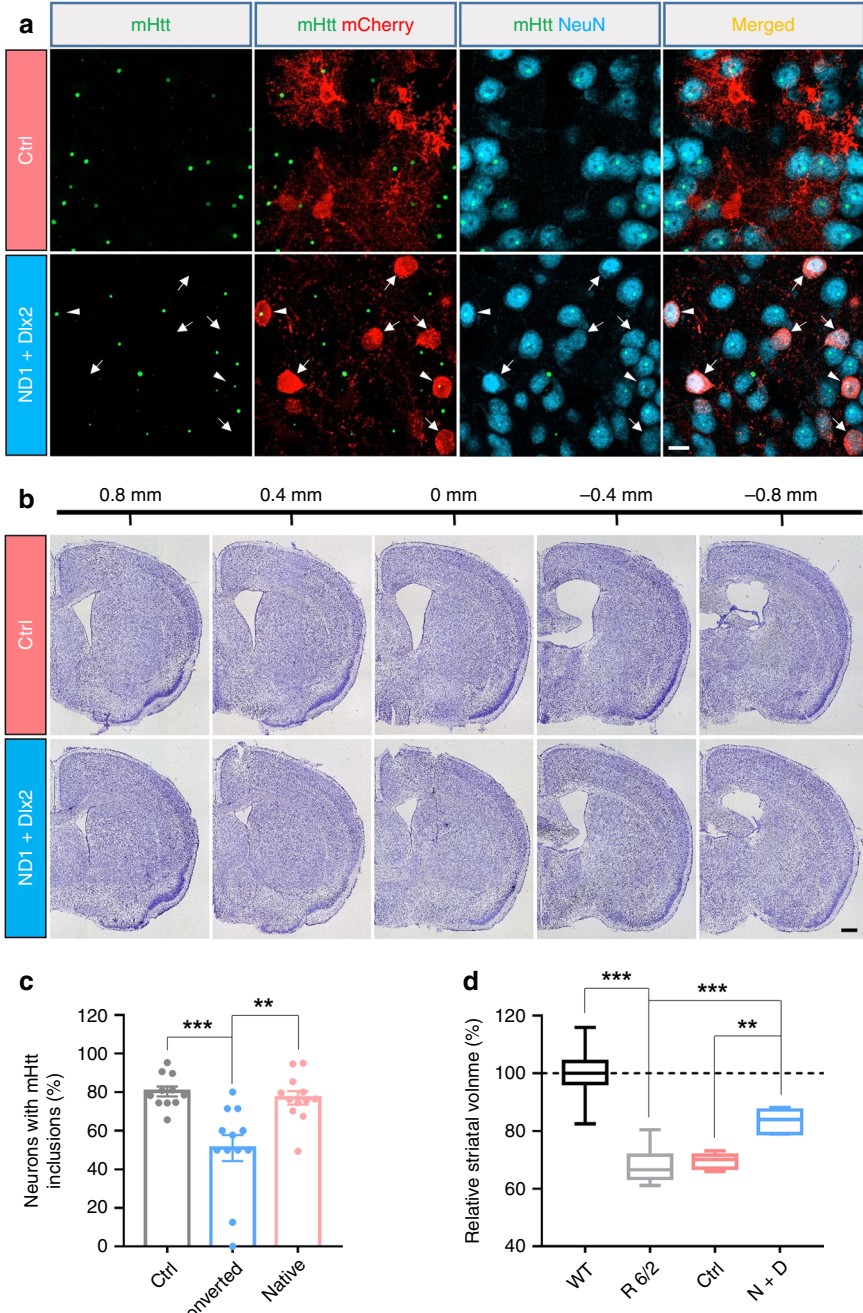

**Fig. 7 Reducing striatum atrophy in the R6/2 mice after in vivo astrocyte-to-neuron conversion. a** Reduction of mHtt inclusions in the striatal astrocyte-converted neurons in the R6/2 mice. The mHtt aggregates (green dots) were detected in most of the striatal neurons (NeuN, cyan), but some NeuroD1 plus Dlx2-converted neurons (red, pointed by arrows) showed no mHtt aggregates. Arrowheads indicate two converted neurons (mCherry+) with mHtt inclusions. Scale bar: 10 μm. **b** Assessing striatum atrophy by Nissl staining of serial coronal sections of the R6/2 mouse brain, treated with control mCherry virus alone (top row) or with NeuroD1 plus Dlx2 AAV (bottom row). Scale bar: 0.5 mm. **c** Quantified data showing that the percentage of neurons with mHtt inclusions in converted neurons was significantly lower compared with their neighboring native neurons or the striatal neurons in the control virus-treated group. Data are shown as mean ± SEM. **d** Summary graphs of the relative striatum volume (normalized to the WT) among R6/2 mice (P90–97), R6/2 mice treated with control viruses, and R6/2 mice treated with NeuroD1 plus Dlx2 viruses. Striatal atrophy was clearly detected in the R6/2 mice (P90–97), but partially rescued by NeuroD1 plus Dlx2 treatment. Data are shown as box plot (boxes, 25-75%; whiskers, 10-90%; lines, median). **p < 0.01, ***p < 0.001, one-way ANOVA with Bonferroni's post hoc test.

The images were acquired by a Zeiss confocal microscope (LSM 800). For quantification, 2–6 regions in the striatum were randomly selected for confocal imaging (×20 lens 2–4 regions; ×40 lens 4–6 regions). Most imaging analysis was performed with Zeiss software ZEN. In order to avoid the impact of human bias on the analysis, some of the mouse information was blinded during confocal imaging. Moreover, image analysis was further performed blindly: the person who did the quantification did not know the injected virus information. After quantification,

another person decoded the mouse information. The Image J software was used for quantifying the intensity of vGAT.

**Electrophysiology.** Brain slices were prepared at 30–32 days after AAV injection, and cut to 300-μm-thick coronal sections with a vibratome (Leica, VTS1200) at room temperature in cutting solution (in mM: 93 NMDG, 93 HCl, 2.5 KCl, 1.25

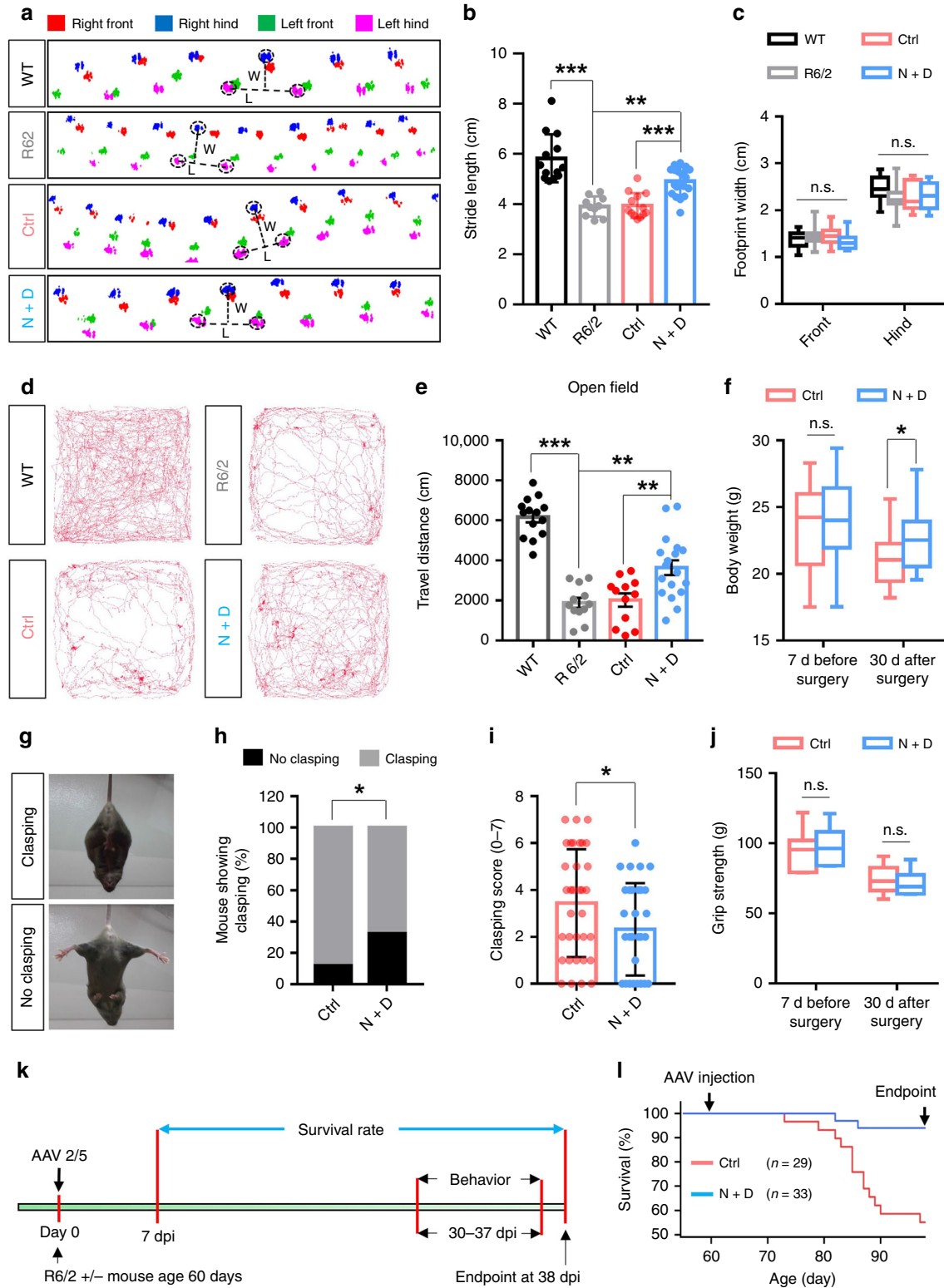

NaH$_2$PO$_4$, 30 NaHCO$_3$, 20 HEPES, 15 glucose, 12 N-Acetyl-L-cysteine, 5 sodium ascorbate, 2 thiourea, 3 sodium pyruvate, 7 MgSO$_4$, 0.5 CaCl$_2$, pH 7.3–7.4, 300 mOsmo, solution was bubbled with 95% O$_2$/5% CO$_2$). Then, slices were transferred to holding solutions with continuous 95% O$_2$/5% CO$_2$ bubbling (in mM): 92 NaCl, 2.5 KCl, 1.25 NaH$_2$PO$_4$, 30 NaHCO$_3$, 20 HEPES, 15 glucose, 12 N-Acetyl-L-cysteine, 5 sodium ascorbate, 2 thiourea, 3 sodium pyruvate, 2 MgSO$_4$, 2 CaCl$_2$. After 0.5–1 h recovery, the slices were transferred to a chamber for electro-physiology study. The recording chamber was filled with artificial cerebral spinal fluid (ACSF) containing: 119 mM NaCl, 2.5 mM KCl, 26 mM NaHCO$_3$, 1.25 mM

NaH$_2$PO$_4$, 2.5 mM CaCl$_2$, 1.3 mM MgCl$_2$ and 10 mM glucose, and constantly bubbled with 95% O$_2$ and 5% CO$_2$ at 32–33 °C. Whole-cell recordings were conducted using a pipette solution consisting of 135 mM K-Gluconate, 5 mM Naphosphocreatine, 10 mM KCl, 2 mM EGTA, 10 mM HEPES, 4 mM MgATP, and 0.5 mM Na$_2$GTP (pH 7.3, adjusted with KOH, 290 mOsm/L). To record the spontaneous synaptic events, the potassium gluconate in the pipette solution was replaced with Cs-methanesulfonate to block K$^+$ channels and reduce noise. Pipette resistance was typically 4–6 MΩ, and series resistance was around 20–40 MΩ. The membrane potential was held at −70 mV for sEPSC recording, and at 0 mV for

**Fig. 8 Functional improvement of the R6/2 mice following in vivo cell conversion. a** Representative footprint tracks among wild-type littermates, R6/2 mice, R6/2 mice treated with control viruses or NeuroD1 + Dlx2 viruses. Dashed lines indicate stride length (L) and width (W). **b, c** Quantified data of stride length (**b**) and width (**c**) among different groups. The stride length decreased in R6/2 mice, but partially rescued by NeuroD1 plus Dlx2 treatment (one-way ANOVA with Bonferroni's post hoc test). **d** Representative tracks showing locomotor activity in the open field test (20 min) among different groups. **e** Quantified data showing the total travel distance reduced in R6/2 mice but partially improved by NeuroD1 plus Dlx2 treatment (one-way ANOVA with Bonferroni's post hoc test). **p < 0.01, ***p < 0.001. **f** Average body weight of R6/2 mice at 7 days before surgery and 30 days after surgery (viral injection). NeuroD1 plus Dlx2-treated R6/2 mice showed less body weight loss than the control virus-treated mice at 30 dpi (*p < 0.05, unpaired Student's t-test). Mouse number in each group is labeled in each bar. **g** Typical clasping (top) and non-clasping (bottom) phenotype in the R6/2 mice. **h** The percentage of mice showing clasping phenotype was decreased in NeuroD1 plus Dlx2-treated R6/2 mice (*p < 0.05, 2-sided Pearson Chi-square test). **i** The average clasping score was also significantly reduced by NeuroD1 plus Dlx2 treatment (*p < 0.05, unpaired Student's t-test). Mouse number in each group is labeled in the bar. **j** The grip strength of R6/2 mice did not change following NeuroD1 plus Dlx2 treatment. **k** Experimental diagram showing survival rate calculation from 7 days post-surgery to 38 days post surgery (endpoint mouse age: P98). Mice that died between 7 and 38 dpi were recorded. Behavioral tests were conducted between 30–37 dpi. **l** Kaplan–Meier survival graph showing that 13 out of 29 R6/2 mice died in the control virus group, whereas only 2 out of 33 R6/2 mice died in the NeuroD1 plus Dlx2 treatment group (p < 0.001, 2-sided Pearson Chi-square test). Data are shown as mean ± SEM in panel **b**, **e** and **i**. Data in panel **ₑc**, **f** and **j** are shown as box plot (boxes, 25-75%; whiskers, 10-90%; lines, median).

## Table 1 Antibodies used in this study.

| Antibodies (dilution) | Host | Source | Catalog # |
|---|---|---|---|
| RFP (1:1000) | Rat mAb | Chromotek | 5f8-100 |
| NeuroD1 (1:1000) | Mouse mAb | Abcam | AB60704 |
| Dlx2 (1:1000) | Rabbit | Abcam | AB30339; discontinued |
| Dlx2 (1:200) | Rabbit | Millipore | AB5726 |
| Cre (1:1000) | Mouse mAb | Millipore | MAB3120 |
| GFAP (1:2000) | Rabbit | Millipore | AB5804 |
| GFAP (1:1000) | Chicken | Millipore | AB5541 |
| Glutamine synthetase (1:1000) | Mouse mAb | Millipore | MAB302 |
| S100β (1:1000) | Rabbit | Abcam | ab52642 |
| NG2 (1:150) | Mouse | Abcam | ab50009 |
| Olig2 (1:1000) | Rabbit | Millipore | AB9610 |
| Iba1 (1:1000) | Rabbit | Wako | 019–19741 |
| NeuN (1:2000) | Guinea Pig | Millipore | ABN90 |
| NeuN (1:2000) | Rabbit | Millipore | ABN78 |
| GAD67 (1:1000) | Mouse mAb | Millipore | MAB5406 |
| GABA (1:1000) | Rabbit | Sigma | A2052 |
| DARPP32 (1:1000) | Rabbit | Millipore | AB10518 |
| Parvalbumin (1:5000) | Mouse mAb | Sigma | P3088 |
| Somatostatin (1:300) | Rat | Millipore | MAB354 |
| NPY (1:2000) | Rabbit | Abcam | AB30914 |
| Calretinin (1:2000) | Goat | Millipore | AB1550 |
| vGAT (1:500) | Guinea Pig | SYSY | 131004 |
| mHtt (1:1000) | Mouse mAb | DSHB | MW7 |
| Ki67 (1:500) | Rabbit | Abcam | Ab15580 |
| S100β (1:1000) | Mouse mAb | Abcam | Ab66028 |

sIPSC recording[50]. Data were collected using pClamp 9 software (Molecular Devices, Palo Alto, CA), sampled at 10 kHz, and filtered at 1 kHz, then analyzed with pClamp 9 Clampfit and MiniAnalysis software (Synaptosoft, Decator, GA).

**Nissl staining and quantification of relative striatum volume**. To assess striatal atrophy, brains were sliced and collected in a serial manner allowing accurate identification of the anterior/posterior sections relative to the bregma so that the striatal volume could be calculated. Every 5th section (anterior and posterior of bregma) covering the entire striatum was included for calculating the striatum volume. Samples were mounted on glass slides and allowed to dry at room temperature for 24 h and then stained with crystal violet. The stained sections were photographed by Keyence microscope (BZ9000). Striatum area was outlined according to the mouse brain atlas and the size of the striatum was blindly measured by Image J software. Striatal volume was calculated using Cavalieri's principle[41] (volume = $s_1d_1 + s_2d_2 + \ldots + s_nd_ns$, s is surface area and d is the distance between two sections). All of the values were normalized to the striatal volume in wild-type littermates.

**Behavioral tests and analyses**. The mice were acclimated to the behavioral testing room for 1 h in order to reduce the effect of the stress associated with movement of the cages. Both female and male mice were included for behavioral tests, and the female and male mouse number was stated in the results section.

Catwalk. The CatWalk XT 10.6 (Noldus) system was used to analyze gait deficits in R6/2 mice. The stride length and footprint width were analyzed to evaluate the treatment effects of in vivo cell conversion. The maximum run duration was 6 s, with a maximum speed variation of 60% in order to reduce variability in the mouse's natural gait pattern. Three compliant trials were acquired per mouse in order to ensure reproducibility. Before each trial the walkway was cleaned with 70% ethanol and dried, then fanned in order to reduce any remaining alcohol odor. During the trial period the room light was turned off. The mouse gait was analyzed automatically by the system software (CatWalk XT 10.6, Noldus). To avoid detections of false footprint, such as mouse excrement, nose-point, tail, and belly, the analysis results were further checked visually and corrected blindly.

Open field test. The open field test was used to assess the locomotion activity in the R6/2 mice. The study arena was a white open-top box ($50 \times 50 \times 30$ cm$^3$), and the mouse was gently placed in the center to start the test. The computer program (EthoVision XT Version 8, Noldus) was calibrated to the arena and set to track center-point, nose-point, and tail-point of the mouse using dynamic subtraction. The mouse freely moved in the open box for 20 min, and its route was automatically tracked and analyzed by the software (Ethovision XT Version 8).

Clasping. The clasping test was used to measure dystonia and dyskinesia[3]. The mouse was suspended upside down by its tail for 14 s. The 14-s trial was split into seven intervals, with 2-s for each interval. The animal was awarded a score of 0 (no clasping) or 1 (clasping). The score for the seven intervals was summed for each mouse allowing a maximum score of 7. Clasping was defined as a behavior whereby paws crossed and came to the chest for any period of time within each 2-s interval[51]. The test was video recorded and analyzed later in a blind fashion.

Mouse weight. The mouse weight was tracked in order to observe any severe weight loss, as the R6/2 mouse model is known to have up to 20% weight decrease after 3 months of age. The mice were weighed individually each Tuesday at 5:00 PM in the animals' homeroom inside an approved vent hood.

Grip strength. The grip strength test was used to quantitatively measure the strength of the mouse forepaws. The grip strength meter (BIO-GS3, Bioseb) was set to record in grams. Each mouse was held by its tail and allowed to grasp the metal grid with only its two front paws. The mouse was pulled until failure to record the maximum strength for each trial. Each mouse was tested three times per time point and the three trials were then averaged to calculate the mean grip strength for each time point tested.

**Statistics**. Data were shown as mean ± standard error of mean (SEM) or mean ± standard deviation (SD), or box plot. Two-tailed Student's t-test (paired or unpaired) was performed to determine the statistical significance between two-group comparison, and the Chi-square test was used to compare the difference of percentage between two groups. One-way ANOVA analysis (GraphPad Prism 7.0) followed by Bonferroni post hoc test was used to for multiple group comparisons. P < 0.05 was considered statistically significant.

**Reporting summary**. Further information on research design is available in the Nature Research Reporting Summary linked to this article.

## Data availability

The data that support the findings of this study are available from the corresponding author upon reasonable request. The source data underlying Figs. 1d, 2c–h, 3e, f, 4c–f,

5f–q, 6c–g, 7c, d, 8b, c, 8e–l, and Supplementary Figs. 3b, c, 5e–g, 6b–d, 7b, 9e–g, 10c, 11b–d, 12d–f, 17b are provided as a Source Data file.

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

## Acknowledgements

This work was supported by NIH grant (AG045656) and Charles H. "Skip" Smith Endowment Fund at Pennsylvania State University to G.C. We thank Yuting Bai for mouse weaning and Mei Jiang for mouse genotyping. We would like to thank all Chen Lab members for scientific discussions throughout the progress of this project.

## Author contributions

Z.W. played a major role in designing and performing most of the experiments, analyzing the data, making figures and writing the initial draft of the paper. G.C. supervised the entire project, designed the experiments with Z.W. and participated in data analysis and revised the paper. M.P., X-Y.H., H.W., R.C., and S.A. performed animal surgery, immunostaining, behavioral test, and some data analysis. M-H.L. performed the in vivo cell conversion studies in YAC128 transgenic mouse model for HD. Z-F.P. prepared all of the AAV2/5 viruses for this study. Y-C.C. made the NeuroD1 and Dlx2 plasmids. Z-Y.G. initiated an early study on GABAergic neuron conversion.

## Competing interests

Gong Chen is a co-founder of NeuExcell Therapeutics Inc.
