## [Peer Review File · Nature Communications]

Reviewers' comments:

Reviewer #1 (Remarks to the Author):

Here the authors pursue a very important topic, namely how to replace degenerating medium spiny neurons in a Huntington Disease mouse model. Towards this aim they transduce astrocytes in the adult murine striatum using either GFAP-driven Cre delivered by AAVs or in a mouse line to revert flexed reporter constructs or constructs containing neurogenic fate determinants. They convincingly demonstrate that they target predominantly astrocytes in the intact brain, and show a gradual decrease in astrocytes and increase in NeuN+ cells when using NeuroD1 and Dlx2. The authors then proceed to use the HD mouse model, and also report the same amount of conversion there. In addition they present convincing electrophysiological analysis and show labelled axon tracts from labelled neurons. Finally, the authors show rescue of striatal atrophy. While this may be an amazing finding, it also raises the following two concerns- i) are there also neurons rescued by the treatment – i.e. fewer neurons degenerate and ii) is the neuron/astrocyte ratio altered. This ought to be examined and possible negative consequences of the latter must be discussed. Finally, the authors present improvement in behavior defects of HD R6/2 mice. Taken together, this is a very exciting study that will be of broad interest to the readers of Nature communications given the concerns raised above and below can be addressed.

Suggestions:

- 1) While the authors depict the number of animals in many histograms, standard deviation and number of animals is missing in Figure 2f. Please add this information and correct the spelling mistake in the labelling of the y-axis.
- 2) For co-localization analysis please show some Z-projections to ensure the actual co-localization rather than 2 cells being on top of each other in the sections. For example in Figure 2b it sometimes appears to be the case...
- 3) Please add standard deviations and dots for each animal in Figure 3 e and f.
- 4) Show Z-projections for some of the high power examples shown in Figure 3c and d.
- 5) I am surprised how 'healthy' the striatum of the R6 HD mouse model looks here (control, Figure 4a). I suggest to add reprogramming experiments at later stages when the disease has progressed further to exclude that many neurons are rescued from the disease and demonstrate that astrocytes still be converted to neurons when a large number of neurons have been lost. An alternative approach would be to quantify total number of neurons not transduced and transduced to see whether also non-transduced neurons increase in number.
- 6) Provide confocal Z-projections for Figures 6e to clearly demonstrate co-localization of CTB labelling, NeuN and mCherry.
- 7) Quantify neuron/astrocyte ration in Ctrl mice and R6/2 mice with reprogramming. If the ratio were altered, is the K-buffering capacity affected?

Reviewer #2 (Remarks to the Author):

Wu et al., describe a de novo conversion of astrocytes into GABAergic neurons using AAV2/5-mediated expression of 2 transcription factors, NeuroD1 and Dlx2 as a treatment approach for Huntington's disease. The authors provide data that suggest that astrocytes are targeted specifically leading to a time-dependent direct conversion of astrocytes into DARPP32+, GABAergic neurons with largely similar electrophysiological properties to native medium spiny neurons (MSNs). The method appears as efficient in a wildtype background as it is in R6/2 HD mice. The authors demonstrate an improvement in several behavioral phenotypes in this mouse model.

The approach described is novel and the study is comprehensive. We do have a few comments that we wish to be addressed by the authors:

Major comments:

1. The data presented on the specificity of rAAV2/5 is not conclusive and it is unclear whether the NeuroD1 and Dlx2 transcription factors would be delivered to other cell-types besides astrocytes. The authors can consider adding a pan Cre control to the experiment described in Figure 3a.
2. Considering the high astrocyte-to-neuron conversion efficiency, the effect of this treatment approach on the astrocyte population in general should be evaluated quantitatively, both in the wildtype and in R6/2 case. It is also important to know if other populations are affected. Stereological analysis of total astrocyte, microglia (Iba1+), and neuronal (NeuN+) densities done at 30dpi in both WT and R6/2 mice would address this point.
3. In Figure 5, the authors compared the electrophysiological properties of native and converted MSNs from R6/2 mice, demonstrating few differences. However, it is difficult to judge the relevance of these differences without a WT control. The authors should address this point.
4. Figure 7 raises an important point that needs to be followed up. Since fewer newly-generated MSNs contain HTT inclusions, do they survive longer than the native MSNs?

Minor comments:

5. What is the rationale for performing the injections on mice from 2-5 months of age in the beginning? Why such a broad age range and is there any change in efficiency of conversion based on the age of the mice?
6. It is unclear whether all the control vs N+D injections from Figures 1-6 were done bilaterally or unilaterally with one side acting as control and the other side N+D. The authors should clarify this in the text.
7. The level of overexpression of the N+D genes in mouse brain over time should be assessed as well as some measures of toxicity/apoptosis.
8. The number of female vs male mice using the behavioural analyses should be provided.
9. For the axonal projection experiments, it would be helpful to show other anatomical positions in the supplementary figures to address the extent of coverage of the treatment/converted MSNs.
10. Is this approach likely to work on human astrocytes? This should be discussed in the manuscript.

Reviewer #3 (Remarks to the Author):

The paper by Chen et al describes an in vivo cell conversion approach to regenerate striatal GABAergic neurons in the R6/2 mice, a mouse model of Huntington disease (HD). The group by Chen et al. used AAV2/5 mediated expression of NeuroD1 and Dlx2 transcriptional factors for gene therapeutic conversion of striatal astrocytes into DARPP32+GABAergic neurons, the latter of which are mostly affected in HD. Authors report up to 80% astrocyte-to-neuron conversion rates and maturation of yielded neurons of more than 50% becoming DARPP32 positive, suggesting successful replenishment of striatal medium spiny neurons (MSN). They confirmed functional properties of gene-therapeutically-converted MSNs by electrophysiological studies, neuronal projection analysis and, most important, improved motor function as well as extension of life span. On the basis of this report, authors deliver proof-of-concept for gene-therapeutic in vivo

conversion of striatal astrocytes into DARPP32+GABAergic neurons representing "a potential disease-modifying therapy to treat HD and other neurodegenerative disorders".

The major claim of the paper is proof of concept for an in vivo cell conversion by gene therapy for the treatment of HD. This is a novel approach of great interest to colleagues in the HD community and other neurodegenerative disorders preferentially those associated with significant cell loss. This conclusion appears to be original. The work is convincing in most aspects, except for the R6/2 mouse model being used, as HD-like phenotype is fulminant, comorbidity (Diabetes) exists in the R6/2 mice, only exon 1 of the mutant Huntingtin sequence is expressed and striatal cell loss is under debate in this mouse of HD. This referee, therefore, would love to see some replication of any findings on cell conversion in a more recent HD-mouse model with higher genetic-construct validity and confirmed striatal cell loss such as 150Q or zQ175DN KI. Nonetheless, overall, this referee feels that the approach and data will influence thinking in the field and that meaningful/appropriate statistics were applied as well as that M&M will enable others to reproduce the work.

We would like to thank all the reviewers for their exciting and constructive comments. We have now collected new data and performed more quantitative analyses to address the excellent questions. We have added line scan Z-projections to better show signal co-localizations. We have analyzed the neuron, astrocyte and microglia density after *in vivo* conversion both in wild type and R6/2 mice. We added brain slice recordings in wild type mice. Most importantly, to further confirm the *in vivo* cell conversion by NeuroD1 and Dlx2 in a different transgenic mouse model of Huntington's disease, we injected AAV NeuroD1 and Dlx2 into the striatum of **15-month old YAC128 mouse model for HD** and observed clear astrocyte-to-neuron conversion (Supplementary Figure 11). We hope the reviewers are satisfied with our revised manuscript and allow our exciting work to be published soon. Thank you.

Point-by-point responses:

Reviewer #1 (Remarks to the Author):

Here the authors pursue a very important topic, namely how to replace degenerating medium spiny neurons in a Huntington Disease mouse model. Towards this aim they transduce astrocytes in the adult murine striatum using either GFAP-driven Cre delivered by AAVs or in a mouse line to revert flexed reporter constructs or constructs containing neurogenic fate determinants. They convincingly demonstrate that they target predominantly astrocytes in the intact brain, and show a gradual decrease in astrocytes and increase in NeuN+ cells when using NeuroD1 and Dlx2. The authors then proceed to use the HD mouse model, and also report the same amount of conversion there. In addition they present convincing electrophysiological analysis and show labelled axon tracts from labelled neurons. Finally, the authors show rescue of striatal atrophy. While this may be an amazing finding, it also raises the following two concerns- i) are there also neurons rescued by the treatment – i.e. fewer neurons degenerate and ii) is the neuron/astrocyte ratio altered. This ought to be examined and possible negative consequences of the latter must be discussed. Finally, the authors present improvement in behavior defects of HD R6/2 mice. Taken together, this is a very exciting study that will be of broad interest to the readers of Nature communications given the concerns raised above and below can be addressed.

Answer: Thank you very much for the very positive comments. As explained below, neuronal protection is indeed a possibility after *in vivo* cell conversion, which has been observed as well in our stroke repair work (Yuchen Chen et al., BioRxiv, 2018). The astrocytes can proliferate after conversion and replenish themselves, as demonstrated in our separate study (Lei Zhang et al., BioRxiv, 2018). In this study, we also further analyzed the neuron/astrocyte ratio and found similar value after conversion (Fig. S5, S6, S9, S10).

Suggestions:

1) While the authors depict the number of animals in many histograms, standard deviation and number of animals is missing in Figure 2f. Please add this information and correct the spelling mistake in the labelling of the y-axis.

Answer: Thanks for the suggestion. The SD is now displayed in the revised Figure 2f and the animal number for each time point is now provided in the figure legend. The spelling mistake is also corrected.

2) For co-localization analysis please show some Z-projections to ensure the actual co-localization rather than 2 cells being on top of each other in the sections. For example in Figure 2b it sometimes appears to be the case...

Answer: Thanks for the suggestion. We have now added the Z-projections in the revised figures. It is clear that the signals are indeed colocalized.

3) Please add standard deviations and dots for each animal in Figure 3 e and f.

Answer: The standard deviations and dots are now added in our revised Figure 3e and f.

4) Show Z-projections for some of the high power examples shown in Figure 3c and d.

Answer: The Z-projections are now added in the revised Figure 3c and d.

5) I am surprised how 'healthy' the striatum of the R6 HD mouse model looks here (control, Figure 4a). I suggest to add reprogramming experiments at later stages when the disease has progressed further to exclude that many neurons are rescued from the disease and demonstrate that astrocytes still be converted to neurons when a large number of neurons have been lost. An alternative approach would be to quantify total number of neurons not transduced and transduced to see whether also non-transduced neurons increase in number.

Answer: Good points. It's our oversight to have focused previously on the extent of AAV-infected area. Now we illustrate images in the R6/2 mouse striatum with more representative image showing enlarged ventricle due to the atrophy of striatum.

The neurodegeneration in R6/2 HD mouse brain was mainly manifested by the tissue atrophy rather than the neuronal density change (see the new Supplementary Figure 9 and 10). When we performed Nissl staining in a series of slices to assess the striatal volume of R6/2 mice, we found that the striatal volume was significantly increased in ND1+Dlx2 treated group than the control group (Figure 7b,d).

For the late stage study, we initially did some reprogramming studies in 3-month old R6/2 mice, but these mice were really sick and none of them can survive more than one week after surgery. Then we moved our study to 2-month old R6/2 mice. Most of the previous studies also intervened at 5-8 weeks old R6/2 mice (Stem Cell Reports. 2018 Jan 9;10(1):58-72; Nature. 2014 May 1;509(7498):96-100; Nat Neurosci. 2014 May;17(5):694-703).

6) Provide confocal Z-projections for Figures 6e to clearly demonstrate co-localization of CTB labelling, NeuN and mCherry.

Answer: Thanks for the suggestion. Z-projections were added in the revised Figure 6e.

7) Quantify neuron/astrocyte ration in Ctrl mice and R6/2 mice with reprogramming. If the ratio were altered, is the K-buffering capacity affected?

Answer: Great point. We have now quantified the neuron/astrocyte ratio in control mice and the R6/2 mice after reprogramming. We have not detected a significant change in the neuron or astrocyte density after in vivo conversion (see new Supplementary Figure 5 and 9). Because astrocyte density did not change significantly, we speculate that the K-buffering capacity might not change too much, but it is worth to study in the future.

Reviewer #2 (Remarks to the Author):

Wu et al., describe a de novo conversion of astrocytes into GABAergic neurons using AAV2/5-mediated expression of 2 transcription factors, NeuroD1 and Dlx2 as a treatment approach for Huntington's disease. The authors provide data that suggest that astrocytes are targeted specifically leading to a time-dependent direct conversion of astrocytes into DARPP32+, GABAergic neurons with largely similar electrophysiological properties to native medium spiny neurons (MSNs). The method appears as efficient in a wildtype background as it is in R6/2 HD mice. The authors demonstrate an improvement in several behavioral phenotypes in this mouse model.

The approach described is novel and the study is comprehensive. We do have a few comments that we wish to be addressed by the authors:

Major comments:

1. The data presented on the specificity of rAAV2/5 is not conclusive and it is unclear whether the NeuroD1 and Dlx2 transcription factors would be delivered to other cell-types besides astrocytes. The authors can consider adding a pan Cre control to the experiment described in Figure 3a.

Answer: This is an important point and has been addressed in our Figure 1. As illustrated, with the use of GFAP::Cre, the Cre expression was indeed targeted in the GFAP-positive astrocytes (Fig. 1b). Fig. 1c,d illustrated that ~90% of our rAAV2/5-infected cells were astrocytes (S100b+, GFAP+, GS+), with very few other types of cells. See quantification of Fig. 1d. Therefore, we conclude that astrocytes are the major target cells infected by our AAV2/5 system due to the restriction of GFAP::Cre. If use a pan-Cre control, it will surely infect other types of cells, not sure that will help specificity.

2. Considering the high astrocyte-to-neuron conversion efficiency, the effect of this treatment approach on the astrocyte population in general should be evaluated quantitatively, both in the wildtype and in R6/2 case. It is also important to know if other populations are affected. Stereological analysis of total astrocyte, microglia (Iba1+), and neuronal (NeuN+) densities done at 30dpi in both WT and R6/2 mice would address this point.

Answer: This is a good point. We have now performed the quantification on astrocyte, microglia, and neuron densities in both WT and R6/2 mice. This information is now presented in our revised manuscript as new Supplementary Figures 5-6 (WT) and 9-10 (R6/2).

3. In Figure 5, the authors compared the electrophysiological properties of native and converted MSNs from R6/2 mice, demonstrating few differences. However, it is difficult to judge the relevance of these differences without a WT control. The authors should address this point.

Answer: Thanks for the suggestion. We have now included additional data on WT brain slice recordings in Figure 5. It appeared that the converted neurons showed slightly lower resting membrane potential and action potential amplitude compared to the WT neurons, whereas the synaptic responses appeared to be similar to the WT neurons.

4. Figure 7 raises an important point that needs to be followed up. Since fewer newly-generated MSNs contain HTT inclusions, do they survive longer than the native MSNs?

Answer: This is an interesting point. With our ND1+Dll2 treatment, our R6/2 mice can survive better than the control group as shown in Fig. 8l. Unfortunately, when we wrote the IACUC protocol, Penn State IACUC required us to sacrifice all the animals around 100 days old, when most of the R6/2 mice would reach the endpoint. We could not tell whether the newly converted neurons would survive better at this time, but we do expect so. Future studies will be designed to answer this important question.

Minor comments:

5. What is the rationale for performing the injections on mice from 2-5 months of age in the beginning? Why such a broad age range and is there any change in efficiency of conversion based on the age of the mice?

Answer: We have not found any obvious difference in terms of conversion efficiency based on the age of the mice (2-5 months). The age of 2-5 months old WT mice used in the beginning was to make sure that we are testing the in vivo astrocyte-to-neuron conversion in the adult mice. Some of the 2 months old WT mice were littermates of the R6/2 mice.

6. It is unclear whether all the control vs N+D injections from Figures 1-6 were done bilaterally or unilaterally with one side acting as control and the other side N+D. The authors should clarify this in the text.

Answer: Thanks for the suggestion. We did bilateral injection of the same virus in R6/2 mice for Figure 4-8. That is: if one R6/2 mouse was injected with control virus, then both sides were injected with control virus.

7. The level of overexpression of the N+D genes in mouse brain over time should be assessed as well as some measures of toxicity/apoptosis.

Answer: This is a good point. As shown in Fig. 2a,b, we have detected both NeuroD1 and Dlx2 expression at 7 dpi in astrocytes (before conversion) and at 30 dpi in newly converted neurons. No obvious difference in terms of expression. We also did not observe obvious toxicity/apoptosis.

8. The number of female vs male mice using the behavioural analyses should be provided.

Answer: Yes, this information was provided in our result section (page 9).

9. For the axonal projection experiments, it would be helpful to show other anatomical positions in the supplementary figures to address the extent of coverage of the treatment/converted MSNs.

Answer: The sagittal section in the Figure 6a illustrate that our newly converted neurons sent axonal projections to globus pallidus (GP) and the substantia nigra pars reticulata (SNr), which is consistent with previous reports (Neuron. 2014 Oct 22;84(2):311-23.; Front Behav Neurosci. 2015 Mar 27;9:71.). The Fig. 6a is now further enlarged in Suppl. Fig. 13 to illustrate that projections to other brain areas was not significant.

10. Is this approach likely to work on human astrocytes? This should be discussed in the manuscript.

Answer: Good point. It seems very difficult to get human striatal astrocytes. While human cortical astrocytes are available, the converted neuron subtype will likely be different due to the different lineages of striatal versus cortical astrocytes. We would like to try it if we can obtain good quality of human striatal astrocytes in the future.

Reviewer #3 (Remarks to the Author):

The paper by Chen et al describes an in vivo cell conversion approach to regenerate striatal GABAergic neurons in the R6/2 mice, a mouse model of Huntington disease (HD). The group by Chen et al. used AAV2/5 mediated expression of NeuroD1 and Dlx2 transcriptional factors for gene therapeutic conversion of striatal astrocytes into DARPP32+GABAergic neurons, the latter of which are mostly affected in HD. Authors report up to 80% astrocyte-to-neuron conversion rates and maturation of yielded neurons of more than 50% becoming DARPP32 positive, suggesting successful replenishment of striatal medium spiny neurons (MSN). They confirmed functional properties of gene-therapeutically-converted MSNs by electrophysiological studies, neuronal projection analysis and, most important, improved motor function as well as extension of life span. On the basis of this report, authors deliver proof-of-concept for gene-therapeutic in vivo conversion of striatal astrocytes into DARPP32+GABAergic neurons representing "a potential disease-modifying therapy to treat HD and other neurodegenerative disorders".

The major claim of the paper is proof of concept for an in vivo cell conversion by gene therapy for the treatment of HD. This is a novel approach of great interest to colleagues in the HD community and other neurodegenerative disorders preferentially those associated with significant cell loss. This conclusion appears to be original. The work is convincing in most aspects, except for the R6/2 mouse model being used, as HD-like phenotype is fulminant, comorbidity (Diabetes) exists in the R6/2 mice, only exon 1 of the mutant Huntingtin sequence is expressed and striatal cell loss is under debate in this mouse of HD. This referee, therefore, would love to see some replication of any

findings on cell conversion in a more recent HD-mouse model with higher genetic-construct validity and confirmed striatal cell loss such as 150Q or zQ175DN KI. Nonetheless, overall, this referee feels that the approach and data will influence thinking in the field and that meaningful/appropriate statistics were applied as well as that M&M will enable others to reproduce the work.

Answer: We thank the reviewer for the very positive comments. In terms of another transgenic mouse model for HD, we are purchasing a new line as suggested and will perform new studies in the future. In the mean time, our collaborators have been using a different transgenic mouse line for HD (YAC128) to repeat our experiments. After consulting with the Editor, here we have included the data obtained from 15-month old YAC128 HD mouse model to demonstrate the astrocyte-to-neuron conversion, see new Supplementary Figure 11.

Reviewers' comments:

Reviewer #1 (Remarks to the Author):

The authors have addressed some of my suggestions, but not others. I have asked to include the dots with the data points from individual animals in Figure 2f, but this is not done. The authors show the SD and give numbers of animals in the legend, but it is important to see the spread of the individual data points - so these should be included. Also the Z-projection now shown in Figure 2b does not show convincing colocalization - there is just a few red dots visible.... if this is the best example the authors have this is worrying. (The example shown in Figure 6, however, is convincing)

My main concern is, however, that they do not find changes in the proportion of astrocytes to neurons if they convert so many astrocytes to neurons. It is correct that astrocytes proliferate in stroke models, but they often fail to proliferate in neurodegenerative models (see e.g. Sirko et al., 2013). This raises concerns that fusion with endogenous neurons may occur and that the authors rather rescue neurons from death than turning glia into neurons. The former would be as interesting and important as conversion, but this would need to be sorted out, e.g. by labelling endogenous neurons.

Reviewer #3 (Remarks to the Author):

The main point raised by this referee "confirm in another HD mouse strain" has been addressed adequately and supportive supplementary data were included. Also, the other points appear to be answered comprehensively.

We appreciate the reviewer's further comments on our gene therapy approach to treat Huntington's disease. We have performed new analysis on the astrocyte proliferation after cell conversion, and confirmed that the proliferative rate significantly increased after *in vivo* astrocyte-to-neuron conversion. The other minor points were also addressed. Hope our exciting work is now ready to be published. Thank you.

Responses to reviewer's comments:

Reviewer #1 (Remarks to the Author):

The authors have addressed some of my suggestions, but not others. I have asked to include the dots with the data points from individual animals in Figure 2f, but this is not done. The authors show the SD and give numbers of animals in the legend, but it is important to see the spread of the individual data points - so these should be included. Also the Z-projection now shown in Figure 2b does not show convincing colocalization - there is just a few red dots visible.... if this is the best example the authors have this is worrying. (The example shown in Figure 6, however, is convincing)

ANSWER: Thanks for the reviewer's suggestion. We have now added the individual data points into our revised Figure 2f. We also enhanced the red signal in revised in Figure 2b so that the colocalization is obvious. Thank you.

My main concern is, however, that they do not find changes in the proportion of astrocytes to neurons if they convert so many astrocytes to neurons. It is correct that astrocytes proliferate in stroke models, but they often fail to proliferate in neurodegenerative models (see e.g. Sirko et al., 2013). This raises concerns that fusion with endogenous neurons may occur and that the authors rather rescue neurons from death than turning glia into neurons. The former would be as interesting and important as conversion, but this would need to be sorted out, e.g. by labelling endogenous neurons.

ANSWER: Astrocyte proliferation in adult brains is correlated to the degree of neuronal injury. Acute injury causes many neuronal death and thus high astrocytic proliferation. In neurodegenerative diseases, neuronal loss is gradual and the rate of astrocytic proliferation is relatively low. In late stage of Alzheimer's disease, we did observe many reactive astrocytes that can be converted by retroviruses expressing NeuroD1 (Guo et al., Cell Stem Cell, 2014). In order to directly test the capability of astrocytic proliferation in R6/2 brains after conversion, we performed Ki67 immunostaining and found that the number of Ki67-labeled dividing astrocytes was remarkably increased in NeuroD1+Dll2 group, compared to the control group. Please check this new data in our new Supplementary Figure 10. Therefore, the endogenous astrocytic proliferation will maintain a relatively unchanged astrocyte to neuron ratio after conversion.

As for fusion between neurons and astrocytes, we have never observed a terminally differentiated neuron fusing with a terminally differentiated astrocyte. Many other groups working on *in vivo* cell conversion also never reported any fusion event between an astrocyte

and a neuron. Our Figure 2 has demonstrated how astrocytes gradually convert into neurons in a time-dependent manner.

Reviewer #3 (Remarks to the Author):

The main point raised by this referee "confirm in another HD mouse strain" has been addressed adequately and supportive supplementary data were included. Also, the other points appear to be answered comprehensively.

ANSWER: Thank you!

REVIEWERS' COMMENTS:

Reviewer #1 (Remarks to the Author):

The authors have now added the individual data points to almost all histograms and show more convincing co-localization in Figure 2b. They also added a New Supplementary Figure 10 to demonstrate proliferation of astrocytes in the reprogramming condition. Unfortunately the authors do not provide any information WHEN this is the case, neither in the Figure legends, nor in the results on p.9. In the paragraph where they describe the data, p. 9, the other data are from 30 days after injection, but I would be very surprised if at this late time point after viral vector injection so many cells would still proliferate. This may be due to the HD model, but apparently this is not the case in the HD model without viral vector injection.

Reviewer 1

The authors have now added the individual data points to almost all histograms and show more convincing co-localization in Figure 2b. They also added a New Supplementary Figure 10 to demonstrate proliferation of astrocytes in the reprogramming condition. Unfortunately the authors do not provide any information WHEN this is the case, neither in the Figure legends, nor in the results on p.9. In the paragraph where they describe the data, p. 9, the other data are from 30 days after injection, but I would be very surprised if at this late time point after viral vector injection so many cells would still proliferate. This may be due to the HD model, but apparently this is not the case in the HD model without viral vector injection.

Answer: We would like to thank the reviewer for the positive comments. Regarding the astroglial proliferation, we performed Ki67 immunostaining at 30 days after AAV injection, because many internal astrocytes infected by ND1+Dlx2 have been converted into neurons at this time (see Figure 2f, bottom). When astrocytes were converted into neurons, the neighboring astrocytes start to proliferate to generate new astrocytes, because astrocytes have the intrinsic property to proliferate and maintain a certain level of gap junctions between astrocytes. We have observed the same phenomenon of astrocyte proliferation after NeuroD1-mediated conversion in the acute brain injury model as well (Zhang et al, ... Chen, BioRxiv, 2018).